# Crosstalks between mTORC1 and mTORC2 variagate cytokine signaling to control NK maturation and effector function

Fangjie Wang[1], Meng Meng[1], Banghui Mo[1], Yao Yang[1], Yan Ji[1], Pei Huang[1], Wenjing Lai[1], Xiaodong Pan[1], Tingting You[1], Hongqin Luo[1], Xiao Guan[1], Yafei Deng[1], Shunzong Yuan[2], Jianhong Chu[3], Michael Namaka[1,4], Tiffany Hughes[5], Lilin Ye[6], Jianhua Yu [5,7], Xiaohui Li [1] & Youcai Deng [1]

The metabolic checkpoint kinase mechanistic/mammalian target of rapamycin (mTOR) regulates natural killer (NK) cell development and function, but the exact underlying mechanisms remain unclear. Here, we show, via conditional deletion of Raptor (mTORC1) or Rictor (mTORC2), that mTORC1 and mTORC2 promote NK cell maturation in a cooperative and non-redundant manner, mainly by controlling the expression of Tbx21 and Eomes. Intriguingly, mTORC1 and mTORC2 regulate cytolytic function in an opposing way, exhibiting promoting and inhibitory effects on the anti-tumor ability and metabolism, respectively. mTORC1 sustains mTORC2 activity by maintaining CD122-mediated IL-15 signaling, whereas mTORC2 represses mTORC1-modulated NK cell effector functions by restraining STAT5-mediated SLC7A5 expression. These positive and negative crosstalks between mTORC1 and mTORC2 signaling thus variegate the magnitudes and kinetics of NK cell activation, and help define a paradigm for the modulation of NK maturation and effector functions.

[1] Institute of Materia Medica, College of Pharmacy, Army Medical University (Third Military Medical University), 30# Gaotanyan Road, Shapingba District, Chongqing 400038, China. [2] Department of Laboratory Medicine, PLA 307 Hospital, Dongdajie 8, Fengtai District, Beijing 100071, China. [3] Institute of Blood and Marrow Transplantation, Soochow University, No. 199 Renai Road, Suzhou 215123, China. [4] Colleges of Pharmacy and Medicine, Rady Faculty of Health Sciences, University of Manitoba, 750 McDermot Avenue, Winnipeg, MB R3E 0T5, Canada. [5] The Ohio State University Comprehensive Cancer Center and the James Cancer Hospital, 460 West 12th Ave, BRT 816, Columbus 43210 OH, USA. [6] Institute of Immunology, Army Medical University (Third Military Medical University), 30# Gaotanyan Road, Shapingba District, Chongqing 400038, China. [7] Division of Hematology, Department of Internal Medicine, The Ohio State University, 460 West 12th Ave, BRT 816, Columbus, OH 43210, USA. Correspondence and requests for materials should be addressed to X.L. (email: lpsh008@aliyun.com) or to Y.D. (email: youcai.deng@tmmu.edu.cn)

Natural killer (NK) cells are a critical component of the innate lymphoid cell subset and function in the immune surveillance of cancers, clearance of virally infected cells, and regulation of the immune system[1, 2]. In particular, the anti-tumor activity of NK cells has been appreciated for decades, and an 11-year follow-up study revealed that people with high-natural cytotoxic activity had a reduced risk of cancer[3]. Thus, harnessing NK cell effector function represents a critical immunotherapeutic approach to cancer and viral infection treatment.

Murine NK cells develop mainly in the bone marrow (BM)[4]. A critical step in murine NK cell differentiation that occurs downstream of the common lymphoid progenitors (CLPs) is the acquisition of the interleukin (IL)-15 receptor β chain (CD122), followed by the expression of NK1.1. After the acquisition of NK1.1, the following three sequential developmental subsets, from immature to mature, can be further classified based on the surface expression of CD11b and CD27: CD11b$^-$CD27$^+$, CD11b$^+$CD27$^+$, and CD11b$^+$CD27$^{-[5, 6]}$. During maturation, NK cells maintain a balance between the expression of activating and inhibitory receptors and can eliminate tumor cells by means of cytotoxic molecules, such as perforin and granzyme B[2]. Additionally, upon activation, NK cells secrete various cytokines, mainly IFN-γ, involved in the regulation of other cell subsets of the immune system[2]. A number of studies have revealed that external factors, such as growth factors and various cytokines (e.g., IL-15, IL-2, and IL-12), as well as intrinsic transcription factors (e.g., Tbx21 and Eomes) are required to control NK cell differentiation, maturation and effector functions[4, 7]. However, links between external factors and intrinsic transcription factors in orchestrating NK cell development and function remain largely unknown.

Mechanistic/mammalian target of rapamycin (mTOR), a highly evolutionarily conserved serine/threonine kinase, acts as a central integrator that regulates anabolic growth and proliferation in response to both extracellular and intracellular signals[8, 9]. mTOR forms the catalytic subunit of two structurally distinct complexes, mTOR complex (mTORC) 1 and mTORC2, that mediate separate but overlapping cellular functions[8]. mTORC1 contains three core proteins, mTOR, Raptor and mLST8, and the central function of these proteins is to direct cell growth and proliferation by regulating anabolic metabolism. Raptor is a regulatory protein associated with mTOR that facilitates recruitment of mTORC1 substrates, including ribosomal protein S6 kinase (S6K) and eukaryotic translation initiation factor 4E binding proteins (4E-BPs), for phosphorylation[8]. mTORC2 also contains mLST8 but uses Rictor (rapamycin insensitive companion of mTOR) instead of Raptor[8]. Rictor is an especially critical adapter protein for mTORC2[8] that can phosphorylate Akt at Ser473, which proves to be the most reliable indicator of mTORC2 activity[10, 11]. Recent findings demonstrated that mTORC1 and mTORC2 direct immune cell fate and function in a highly context-specific manner due to the effects influenced by the developmental stages of immune cells or environmental cues[9]. Recent studies involving rapamycin treatment or mTOR deletion indicate that the kinase mTOR controls a key checkpoint in NK cell differentiation and activation that occurs downstream of IL-15 and requires a negative signal from Tsc1[12, 13]. However, how mTOR signaling mediates these cellular functions, especially how mTORC2 and its cooperation with mTORC1 control NK cell development and effector function, remains unclear. In addition, how mTOR interacts with key transcriptional factors responsible for NK cell development and effector functions also remains largely unknown.

Here, we show that mTORC1 and mTORC2 control NK cell homeostasis and maturation in a cooperative and nonredundant manner while playing a positive or negative role, respectively, in the regulation of NK cell antitumor ability and metabolism. Furthermore, we demonstrate that mTORC1 sustains mTORC2 activity by maintaining CD122-mediated IL-15 signaling, whereas mTORC2 represses mTORC1 by controlling NK cell effector functions mainly through restraining STAT5-mediated SLC7A5 expression. These findings bring new insights regarding the interplay that occurs among mTOR and STAT5 signaling in NK cells.

## Results

**mTORC2 promotes NK cell specification and maturation.** To explore the role of mTORC2 in NK cells, we generated hematopoietic-specific Rictor-deleted mice and NK-specific Rictor knockout mice by crossing mice carrying floxed *Rictor* alleles (*Rictor*$^{fl/fl}$) with *Vav1*-Cre mice or *Ncr1*-Cre mice, respectively. Hematopoietic-specific Rictor-deleted mice (*Rictor*$^{fl/fl}$/*Vav1*-Cre$^+$) will be referred to as Rictor$^{Δ/Δ}$ mice, in which Rictor expression was deleted before lymphoid specification, including hematopoietic stem cells, CLPs, and NK cell precursors as well as mature NK cells. In contrast, NK-specific Rictor knockout mice (*Rictor*$^{fl/fl}$/*Ncr1*-Cre$^+$) will be referred to as Rictor$^{ΔNK}$ mice, in which *Rictor* was deleted during the terminal stage of NK cell development after the acquisition of NKp46. *Rictor*$^{fl/fl}$ mice were used as controls for the Rictor$^{Δ/Δ}$ mice, while mice with a single allele of *Rictor* and *Ncr1*-Cre$^+$ (*Rictor*$^{fl/+}$/*Ncr1*-Cre$^+$) were set as controls for the Rictor$^{ΔNK}$ mice. The efficiency of Rictor deletion and mTORC2 activity downregulation (indicated by the level of Akt phosphorylation at Ser473 (p-Akt$^{Ser473}$)) was confirmed in both the Rictor$^{Δ/Δ}$ and Rictor$^{ΔNK}$ mice via flow cytometry (Supplementary Fig. 1a–c). Initial flow cytometric analysis showed that the relative ratio and absolute quantity of CD3$^-$CD19$^-$CD122$^+$ cells were robustly decreased in the BM of the Rictor$^{Δ/Δ}$ mice. Among CD3$^-$CD19$^-$CD122$^+$ cells, mTORC2 deficiency resulted in a slightly increased ratio of NK1.1$^-$NKp46$^-$ cells without affecting the ratio of NK1.1$^+$NKp46$^-$ and NK1.1$^+$NKp46$^+$ cells in the Rictor$^{Δ/Δ}$ mice (Fig. 1a). We next determined the relative ratio and absolute quantity of NK cells (CD3$^-$CD19$^-$NK1.1$^+$) in the Rictor$^{Δ/Δ}$ mice. The relative ratio of NK cells was reduced in the spleen but unaltered in the BM and lymph nodes, while the absolute quantity of NK cells (CD3$^-$CD19$^-$NK1.1$^+$) was dramatically reduced in all organs of the Rictor$^{Δ/Δ}$ mice (Fig. 1b). As such, these findings were indicative of impaired NK cell differentiation due to Rictor deletion before lymphoid specification.

Unlike the Rictor$^{Δ/Δ}$ mice, the relative ratio and absolute quantity of CD3$^-$CD19$^-$CD122$^+$ cells and the ratio of NK1.1$^-$NKp46$^-$, NK1.1$^+$NKp46$^-$ and NK1.1$^+$NKp46$^+$ cells among CD3$^-$CD19$^-$CD122$^+$ cells were unaltered in the Rictor$^{ΔNK}$ mice (Fig. 1c). The relative ratio and absolute quantity of NK cells (CD3$^-$CD19$^-$NK1.1$^+$) were consistently reduced in the spleen but not in the BM or lymph nodes of the Rictor$^{ΔNK}$ mice (Fig. 1d). Together, these data reveal that mTORC2 activity is required for NK cell specification and distribution.

We next investigated the role of mTORC2 activity in NK cell maturation in both the Rictor$^{Δ/Δ}$ and Rictor$^{ΔNK}$ mice. The data revealed a reduced proportion of CD11b$^+$CD27$^-$ NK cells but an increased proportion of both CD11b$^-$CD27$^+$ and CD11b$^+$CD27$^+$ NK cells in the BM, spleen, and lymph nodes of the Rictor$^{Δ/Δ}$ mice compared with those in the control mice (Fig. 1e). Accordingly, a strong bias toward the over representation of CD27$^+$ cells over CD27$^-$ cells among CD11b$^+$ NK cells was also observed in all the aforementioned organs of the Rictor$^{Δ/Δ}$ mice (Supplementary Fig. 1d). In addition, NK cell maturation in the Rictor$^{ΔNK}$ mice was changed in a manner similar, albeit to a lesser extent, to that seen in the Rictor$^{Δ/Δ}$ mice (Fig. 1f and Supplementary Fig. 1e).

During the final stages of maturation, NK cells gradually upregulate CD11b surface expression while downregulating CD27 expression and finally acquire coexpression of CD43 and

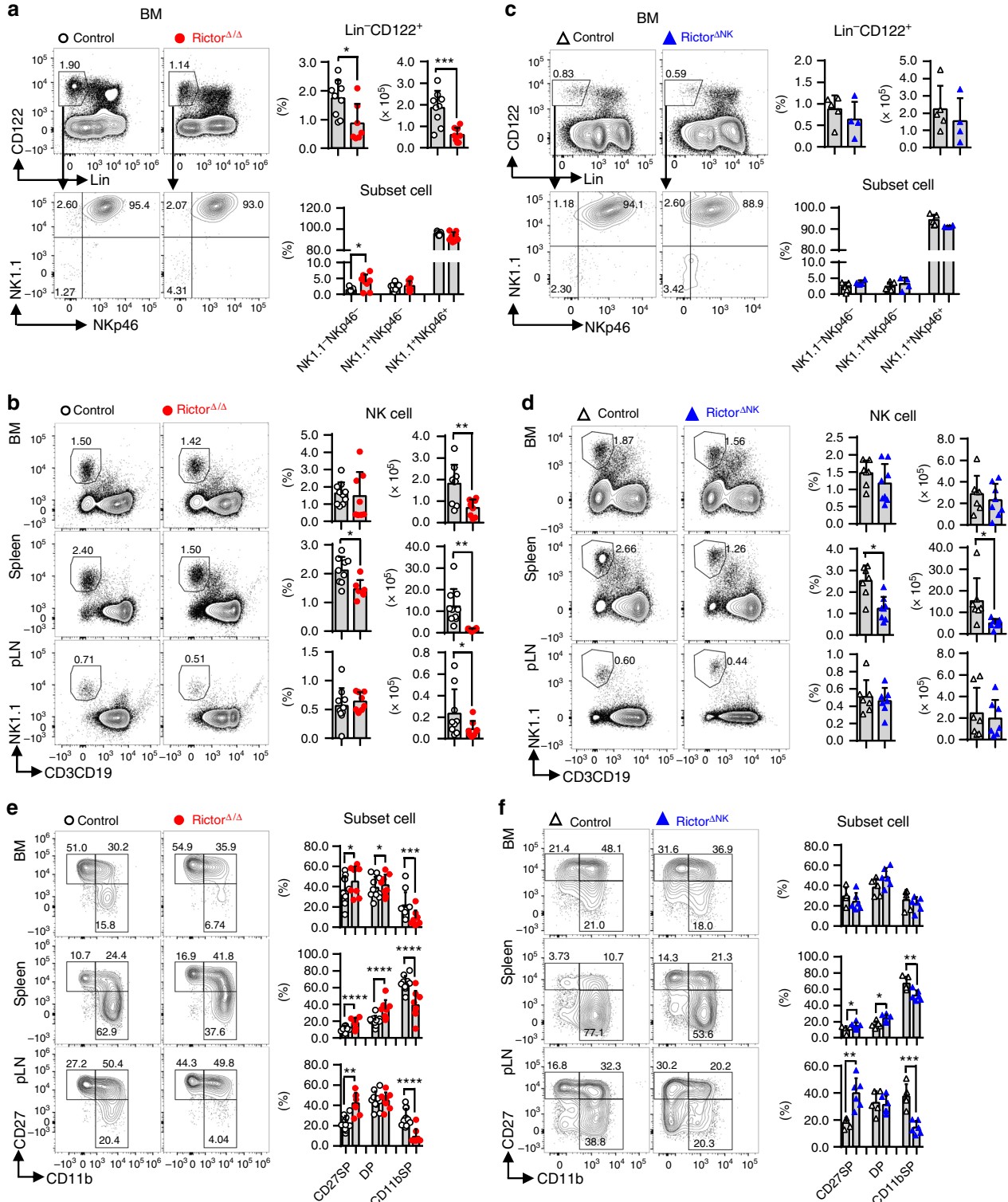

**Fig. 1** Inhibition of mTORC2 activity results in impaired NK cell specification and maturation. **a**, **c** Flow cytometric analysis and enumeration of Lin⁻CD122⁺ (Lin: CD3, CD19, Ter119, Gr1) BM cells (top) and percentage of Lin⁻CD122⁺ BM cells expressing NK1.1 and/or NKp46 (bottom) from control (*Rictor*^fl/fl^/ *Vav1*-Cre⁻) versus Rictor^Δ/Δ (*Rictor*^fl/fl^/*Vav1*-Cre⁺) (**a**) or control (*Rictor*^fl/+^/*Ncr1*-Cre⁺) versus Rictor^ΔNK (*Rictor*^fl/fl^/*Ncr1*-Cre⁺) mice (**c**). **b**, **d** Flow cytometric analysis and enumeration of NK cells (CD3⁻CD19⁻NK1.1⁺) in BM, spleen, and peripheral lymph nodes (pLNs) from control versus Rictor^Δ/Δ (**b**) or control versus Rictor^ΔNK mice (**d**). **e**, **f** Flow cytometric analysis and cumulative frequencies of subpopulations of NK cells (CD3⁻CD19⁻NK1.1 ⁺NKp46⁺) in the BM, spleen, and pLNs from control versus Rictor^Δ/Δ (**e**) or control versus Rictor^ΔNK (**f**) mice. CD27SP, DP, and CD11bSP represent CD27 ⁺CD11b⁻, CD27⁺CD11b⁺, and CD27⁻CD11b⁺ NK cell subsets, respectively. For the bar graphs, each dot represents one mouse, and all experiments were replicated 3 (**c**), 4 (**f**), or 5 (**a**, **b**, **d**, **e**) times; error bars represent SD; *p < 0.05, **p < 0.01, ***p < 0.001, and ****p < 0.0001; unpaired two-tailed Student's *t* test with Welsh's correction (**a**–**f**)

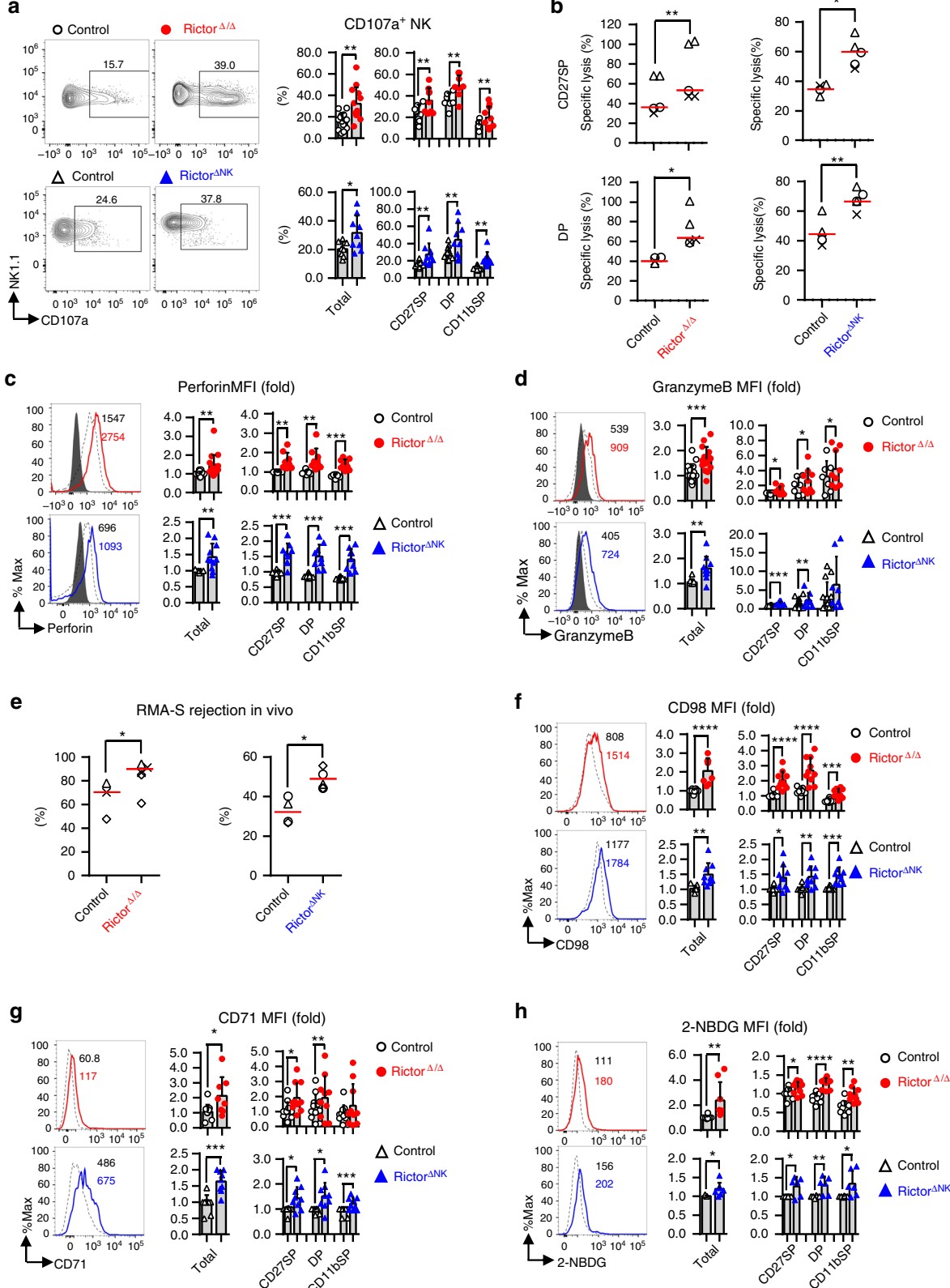

KLRG1[14]. The ratio of CD43[+] KLRG1[+] NK cells from both the Rictor[Δ/Δ] and Rictor[ΔNK] mice was significantly reduced in the BM, spleen, and lymph nodes (Supplementary Fig. 1f, g). The more pronounced increase in immature NK cells and decrease in

mature NK cells seen in the Rictor[Δ/Δ] mice compared with those in the Rictor[ΔNK] mice suggests that Rictor deficiency blocks NK cell maturation at an earlier stage, likely soon after NK cells acquire CD27 expression.

**Fig. 2** mTORC2 negatively regulates NK cell cytotoxicity against tumor cells. **a** Flow cytometric analysis and cumulative frequencies of CD107a$^+$ cells among total splenic NK cells (CD3$^-$CD19$^-$NK1.1$^+$NKp46$^+$) and within the indicated NK cell subpopulations from control versus Rictor$^{\Delta/\Delta}$ (top) or control versus Rictor$^{\Delta NK}$ (bottom) mice after co-culture with Yac-1 targets in the presence of GolgiStop for 6 h. **b** The specific lysis of RMA-S tumor cells when co-cultured with NK cell subsets from control *versus* Rictor$^{\Delta/\Delta}$ or control versus Rictor$^{\Delta NK}$ mice for 4 h at an E:T ratio of 1:1; NK cell subsets were sorted based on the expression of CD27 and CD11b, while RMA-S cells were labeled with Calcein-AM. **c, d** Flow cytometric analysis depicting perforin (**c**) and granzyme B (**d**) expression by splenic NK cells (CD3$^-$CD19$^-$NK1.1$^+$NKp46$^+$) and subpopulations thereof from control versus Rictor$^{\Delta/\Delta}$ (top) or control *versus* Rictor$^{\Delta NK}$ mice (bottom) mice. **e** The cumulative ratio of RMA-S cell rejection 18 h after intraperitoneal injection of a 1:1 mixture of Cell Trace Far Red-labeled RMA and Cell Trace Violet-labeled RMA-S cells into control versus Rictor$^{\Delta/\Delta}$ or control versus Rictor$^{\Delta NK}$ mice. **f–h** Flow cytometric analysis and cumulative results depicting expression of CD98 (**f**), CD71 (**g**), and the uptake of 2-NBDG (**h**) by total splenic NK cells or the indicated NK cell subsets, comparing control versus Rictor$^{\Delta/\Delta}$ (top) or control versus Rictor$^{\Delta NK}$ (bottom) mice. The MFI was calculated relative to total NK cells or the CD27SP NK cell subset from control littermates. CD27SP, DP, and CD11bSP represent CD27$^+$CD11b$^-$, CD27$^+$CD11b$^+$, and CD27$^-$CD11b$^+$ NK cell subsets, respectively. For the bar graphs, each dot represents one mouse, and all experiments were replicated 5 (h bottom), 6 (a bottom, f, g, h top), 7 (c bottom, d bottom), or 8 (a top, c top, d top) times; error bars represent SD; *$p < 0.05$, **$p < 0.01$, ***$p < 0.001$, and ****$p < 0.0001$; unpaired two-tailed Student's *t* test with Welsh's correction (**a, c, d, f–h**); generalized linear models (**b, e**). Each of the shapes represents a matched pair of littermates in an individual experiment, and the horizontal red line represents the median (**b, e**)

**mTORC2 negatively regulates NK cell cytotoxicity**. We next explored whether mTORC2 deficiency would also affect NK cell effector functions. We first tested whether mTORC2 deficiency affected NK cell responsiveness to stimulation with PMA and ionomycin. The overall proportion and mean fluorescence intensity (MFI) of IFN-γ-secreting NK cells were unchanged following 6 h of stimulation with PMA and ionomycin in both the Rictor$^{\Delta/\Delta}$ and Rictor$^{\Delta NK}$ mice (Supplementary Fig. 2a, b).

We then tested the cytotoxic potential of Rictor-deficient NK cells. Surface expression of CD107a, a surrogate marker used to measure NK cell degranulation following stimulation[15], was increased in each developmental subset of NK cells identified based on surface expression of CD11b and CD27 from both the Rictor$^{\Delta/\Delta}$ and Rictor$^{\Delta NK}$ mice compared with that in NK cells from the control mice, indicating an enhanced "per-cell" cytotoxic potential of Rictor-deficient NK cells against Yac-1 target cells (Fig. 2a). Then, directly ex vivo, both CD11b$^-$CD27$^+$ and CD11b$^+$CD27$^+$ NK cells sorted from both Rictor$^{\Delta/\Delta}$ and Rictor$^{\Delta NK}$ mice possessed stronger ability to lyse RMA-S cells than the corresponding subsets of NK cells from WT mice (Fig. 2b). Consistently, a significant increase in basal levels of classic NK cell cytotoxic molecules, such as perforin and granzyme B, was also found in NK cells from both the Rictor$^{\Delta/\Delta}$ and Rictor$^{\Delta NK}$ mice (Fig. 2c, d). These findings prompted us to further test whether mTORC2 deficiency in NK cells affects in vivo antitumor activity. We used RMA-S tumor cells, which lack MHC class I expression and specifically activate NK cells in vivo[16], combined with the NK-resistant RMA cells to evaluate the specific clearance of RMA-S cells by NK cells. Unexpectedly, in contrast with the results of more immature NK cells, mTORC2 deficiency increased the efficiency of NK cell rejection of RMA-S cells in both the Rictor$^{\Delta/\Delta}$ and Rictor$^{\Delta NK}$ mice compared with that in the wild-type (WT) controls (Fig. 2e).

Cell metabolism is a basic process critical for lymphocyte differentiation and regulates the immune response[17, 18]. Over the course of 18–24 h, cytokine stimulation activates NK cells, which undergo metabolic reprograming that increases the rate of both glycolysis and mitochondrial oxidative phosphorylation (OXPHOS)[19–21]; a similar, though less rapid, process reportedly occurs in T cells[22]. Because mTOR acts as a metabolic checkpoint kinase, we next tested whether mTORC2 deficiency interfered with the cellular metabolic status of NK cells. The expression levels of dedicated transporters, including transferrin receptor CD71 and the amino-acid transporter CD98, controlling cell access to nutrients[17, 18], were substantially increased on the surface of NK cells from the Rictor$^{\Delta/\Delta}$ mice (Fig. 2f, g). Glucose uptake, measured via uptake of the fluorescent glucose analog 2-(N-(7-nitrobenz-2-oxa-1,3-diazol-4-yl) amino)-2-deoxyglucose (2-NBDG)[12, 23], was also increased in NK cells from the Rictor$^{\Delta/\Delta}$ mice (Fig. 2h). Moreover, NK cells from the Rictor$^{\Delta NK}$ and Rictor$^{\Delta/\Delta}$ mice showed similar trends in CD71 and CD98 surface expression as well as in 2-NBDG uptake (Fig. 2f±h). Taken together, these findings suggest an increased overall rate of cellular metabolism in Rictor-deficient NK cells.

Mitochondria, the essential hubs of metabolic activity, are pivotal for OXPHOS activity and immune responses[24] and have recently been shown to be regulated by mTOR signaling[25, 26]. In general, healthy mitochondria generate a proper membrane potential for importing proteins or substrates from the cytosol into the mitochondrial matrix for OXPHOS accompanied by reactive oxygen species (ROS) generation[27]. We found that NK cells from the Rictor$^{\Delta/\Delta}$ mice had increased levels of mitochondria-associated ROS and overall mitochondrial content and that the mitochondria exhibited an increased mitochondrial membrane potential, as indicated by flow cytometric labeling with tetramethylrhodamine ethyl ester (TMRE), MitoSox, and Mito-Tracker (Supplementary Fig. 2c). We also found that the ROS content on a per-mitochondrion basis, as calculated by the ratio of MitoSox/MitoTracker, was normal in NK cells from the Rictor$^{\Delta/\Delta}$ mice (Supplementary Fig. 2d), suggesting a relatively normal capacity for ROS clearance in these NK cells. Additionally, the decrease in per-mitochondrion membrane potential, as calculated by the ratio of TMRE/Mitotracker, in the Rictor$^{\Delta/\Delta}$ NK cells was consistent with our observation of enhanced antitumor capacity and was also in agreement with previous reports in CD8$^+$ T cells[28] (Supplementary Fig. 2d).

Taken together, these data demonstrate an unexpected role for mTORC2, opposing its role in promoting NK cell maturation. Specifically, mTORC2 appears to inhibit the direct cytotoxic effector function of NK cells in a cell intrinsic manner, potentially by constraining cellular metabolism, for example, by reducing nutrient uptake and "per-cell" mitochondrial activity.

**NK cell development and effector functions require mTORC1**. Previous studies have reported that repressing mTORC1 activity via rapamycin treatment inhibits NK cell effector function[12, 19], while heightening mTORC1 activity through an NK cell-specific Tsc1 deletion does not increase[13] NK cell effector function. To determine the exact role of mTORC1 in regulating NK cell development and effector function in a more physiological context, we genetically deleted the core protein of mTORC1, Raptor[8], specifically in NK cells by crossing Ncr1-Cre mice with mice possessing a loxP-flanked Rptor allele (Rptor$^{fl/fl}$/Ncr1-Cre$^+$, herein referred to as Raptor$^{\Delta NK}$ mice). Indeed, splenic NK cells from the Raptor$^{\Delta NK}$ mice showed high efficient Raptor deletion and had decreased levels of phosphorylated (p)-S6$^{Ser235/236}$, a critical marker of mTORC1 activity[8] (Supplementary Fig. 3a–c).

However, to our surprise, in contrast with Rictor or mTOR-deficient NK cells[12], the relative ratio and absolute quantity of CD3−CD19−CD122+ cells and the ratio of NK1.1+NKp46+ cells among CD3−CD19−CD122+ cells were significantly increased in the BM of the Raptor[ΔNK] mice (Supplementary Fig. 3c). Although the relative ratio and absolute quantity of NK cells (CD3−CD19−NK1.1+) from the Raptor[ΔNK] mice were significantly increased in the BM, these metrics both remained unchanged in the spleen, while they were decreased in the lymph nodes (Fig. 3a). This

phenomenon is likely attributable to an increased rate of proliferation with a simultaneous reduction in apoptosis of NK cells in the BM but not in the spleen of the Raptor[ΔNK] mice (Supplementary Fig. 3e, f).

Regarding NK cell maturation, Raptor deficiency decreased the proportion of CD11b+CD27− NK cells while increasing the proportion of CD11b−CD27+ NK cells, concomitant with the reduced ratio of CD27− cells to CD27+ cells among CD11b+ NK cells in the BM, spleen, and peripheral lymphoid nodes (Fig. 3b

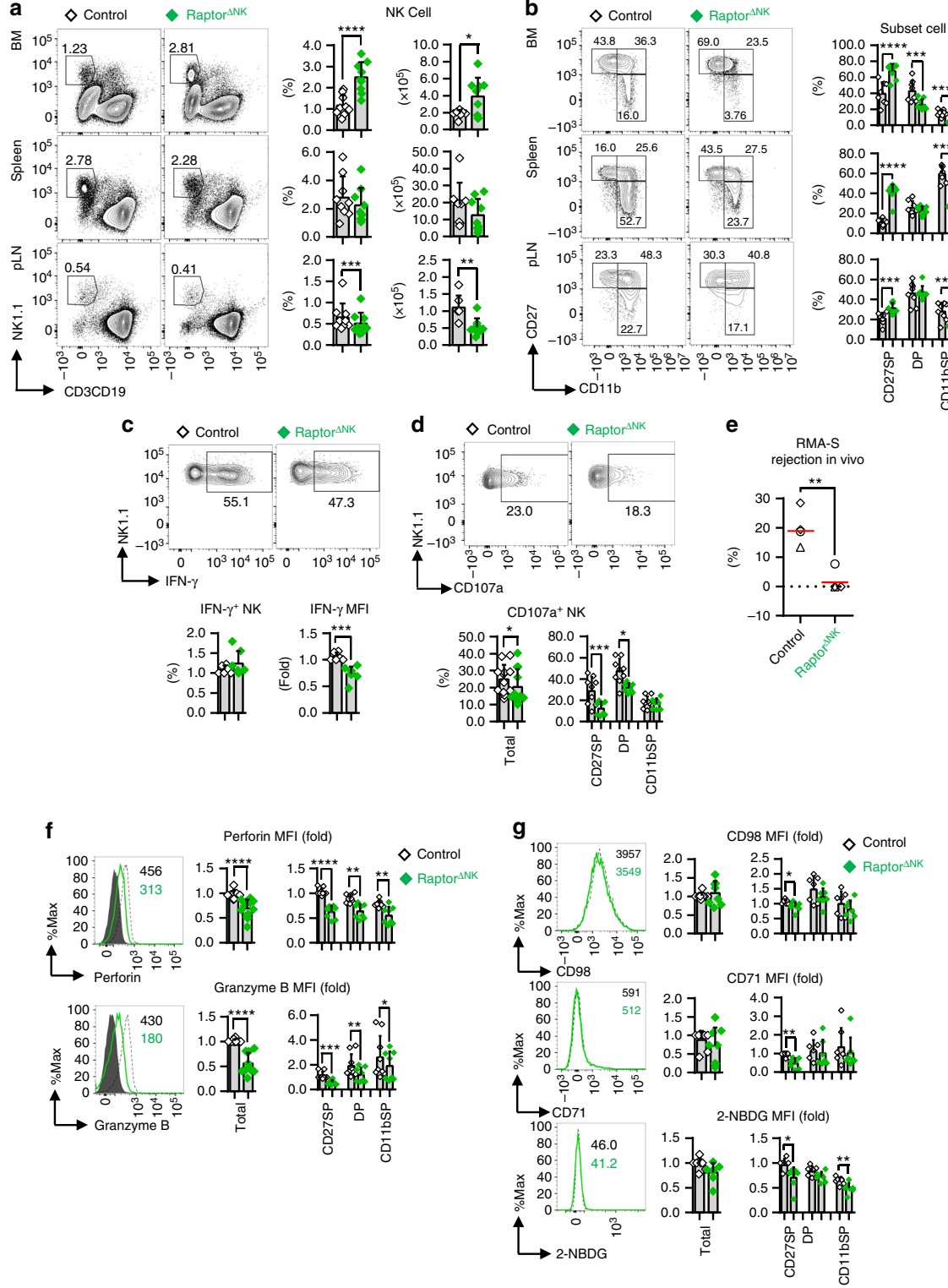

and Supplementary Fig. 3g). These findings together are indicative of mTORC1 promotion of maturation from CD11b$^-$CD27$^+$ to CD11b$^+$CD27$^+$ NK cells. A robust reduction in the ratio of CD43$^+$KLRG1$^+$ NK cells was also observed in the Raptor$^{\Delta NK}$ mice (Supplementary Fig. 3g). Taken together, these findings suggest that Raptor deficiency blocks NK cell maturation after they acquire CD11b expression.

We next analyzed splenic Raptor$^{\Delta NK}$ NK cells for expression of a broad panel of markers responsible for activation and effector functions. In response to PMA and ionomycin stimulation, the overall proportion of splenic Raptor-deficient NK cells secreting IFN-γ was unaltered. However, IFN-γ expression was reduced on a "per-cell" basis, as indicated by the MFI of IFN-γ-secreting cells (Fig. 3c). This seems to be inconsistent with the finding that 20 nM rapamycin significantly reduced the proportion of IFN-γ$^+$ NK cells[19], which might because of the relative higher efficiency of mTORC1 inhibition by rapamycin than our genetically Rptor knockout model. When challenged with YAC-1 target cells, the expression of CD107a was decreased on total NK cells and within the indicated NK subsets in the Raptor$^{\Delta NK}$ mice (Fig. 3d). Regarding the in vivo cytotoxicity, the rate of rejection of RMA-S cells in vivo was substantially reduced in the Raptor$^{\Delta NK}$ mice (Fig. 3e). Consistently, basal levels of perforin and granzyme B expression, CD98 and CD71 expression, and uptake of 2-NBDG were all significantly diminished within some subsets of splenic Raptor-deficient NK cells (Fig. 3f, g).

Taken together, these data reveal that mTORC1 is a critical cell-intrinsic regulator of NK cell specification, maturation, and effector functions; furthermore, mTORC1 seems to prevent NK cell overpopulation in the BM by limiting their proliferation and survival.

**Crosstalk between mTORC1 and mTORC2 in NK cells.** The above data suggest that mTORC1 and mTORC2 both positively regulate NK cell maturation; however, mTORC1 promotes while mTORC2 seems to inhibit NK cell cytolytic functions. These interesting findings led us to further explore whether and how mTORC1 and mTORC2 influence one another's activity, specifically with respect to their regulation of NK cell development and effector functions. Previous studies have shown that cytokines, especially IL-15, are chief modulators of the extracellular signaling that results in the activation of mTOR signaling[12]. Therefore, we used a human NK cell line, NKL cells, and mouse splenic NK cells to profile the activation states of both mTORC1 and mTORC2 during IL-15-mediated NK cell stimulation, which is critical for NK cell survival, proliferation, and activation[29–31]. Consistent with previous findings in murine NK cells[12], the stimulation of NK cells with IL-15 for as little as 1 h led to a rapid and robust increase in the levels of both p-S6$^{Ser235/236}$ and p-

Akt$^{Ser473}$, which were both gradually downregulated in both NKL cells and splenic murine NK cells 12 h later, irrespective of whether the cells were treated with a high (100 ng/ml) or low concentration (10 ng/ml) of IL-15 (Fig. 4a, b). Next, we assessed mTORC2 or mTORC1 activity in mTORC1-deficient (Raptor$^{\Delta NK}$ mice) or mTORC2-deficient (Rictor$^{\Delta/\Delta}$ and Rictor$^{\Delta NK}$ mice) NK cells, respectively. Raptor-deficient NK cells showed reduced mTORC2 activity, as reflected by decreased levels of p-Akt$^{Ser473}$ (Fig. 4c). This phenomenon may be attributable to a reduction in IL-15 responsiveness because decreased expression of the IL-15 receptor CD122 was also observed on NK cells from the Raptor$^{\Delta NK}$ mice (Fig. 4d). Accordingly, phosphorylation of STAT5 (p-STAT5), which is another critical mediator that occurs downstream of IL-15 signaling[32], was also reduced (Fig. 4e).

In contrast, Rictor deficiency led to a nearly two fold increase in S6 phosphorylation (p-S6$^{Ser235/236}$) in total and in each subset of NK cells from both the Rictor$^{\Delta/\Delta}$ and Rictor$^{\Delta NK}$ mice (Fig. 4f). Following ex vivo stimulation with IL-15, levels of p-S6$^{Ser235/236}$ in Rictor-deficient NK cells were increased by approximately eightfold compared with less than a threefold increase in control NK cells (Fig. 4g).

Taken together, these data indicate that Raptor-mediated mTORC1 activity is critical for maintaining mTORC2 activity by regulating IL-15 signaling, while Rictor-mediated mTORC2 activity might represent a negative feedback regulator of Raptor-mediated mTORC1 activation, guaranteeing the proper magnitude of NK cell development and activation by counteracting mTORC1 hyperactivation.

**mTORC2 and mTORC1 nonredundantly promote NK cell maturation.** To further confirm our above hypothesis that mTORC2 might represent a negative feedback mechanism to modulate mTORC1-mediated regulation of NK cell homeostasis, maturation, and effector functions, we utilized a genetic approach to attenuate the heightened levels of mTORC1 activity observed in Rictor-deficient NK cells by deleting a single Rptor allele in the Rictor$^{\Delta NK}$ mice. For this purpose, we first crossed Rictor$^{fl/fl}$ mice with mice carrying floxed Rptor alleles (Rptor$^{fl/fl}$) to generate Rictor$^{fl/+}$Rptor$^{fl/+}$ mice and then crossed Rictor$^{fl/fl}$/Ncr1-Cre$^+$ (Rictor$^{\Delta NK}$) mice with Rictor$^{fl/+}$Rptor$^{fl/+}$ mice to generate control (Rictor$^{fl/+}$/Ncr1-Cre$^+$), Rictor$^{\Delta NK}$ (Rictor$^{fl/fl}$/Rptor$^{+/+}$/Ncr1-Cre$^+$), and Rictor$^{\Delta NK}$Rptor$^{fl/+}$ (Rictor$^{fl/fl}$/Rptor$^{fl/+}$/Ncr1-Cre$^+$) mice. Intracellular flow cytometric analysis revealed that NK cells from the Rictor$^{\Delta NK}$ Raptor$^{fl/+}$ mice recovered levels of mTORC1 activity comparable to those in control mice, while levels of mTORC2 activity were comparable to those in Rictor-deficient NK cells (Fig. 5a). However, in contrast to our expectations, the recovery of mTORC1 activity in the Rictor$^{\Delta NK}$Raptor$^{fl/+}$ mice was unable to rescue and, in fact, further intensified the decrease

**Fig. 3** NK cell specification, maturation, and effector functions require mTORC1 activity. **a** Cumulative ratio and enumeration of NK cells (CD3$^-$CD19$^-$NK1.1$^+$) present in BM, spleen, and peripheral lymph nodes (pLNs) from control (Rptor$^{fl/+}$/Ncr1-Cre$^+$) versus Raptor$^{\Delta NK}$ (Rptor$^{fl/fl}$/Ncr1-Cre$^+$) mice. **b** Flow cytometric analysis and cumulative frequencies of NK cell (CD3$^-$CD19$^-$NK1.1$^+$NKp46$^+$) subsets found in the BM, spleen, and pLNs from control versus Raptor$^{\Delta NK}$ mice. **c** Statistical quantification of IFN-γ$^+$ splenic NK cells (CD3$^-$CD19$^-$NK1.1$^+$NKp46$^+$) and the MFI of IFN-γ among IFN-γ$^+$ NK cells following stimulation with PMA and ionomycin in the presence of GolgiPlug for 6 h. **d** Flow cytometric analysis and cumulative frequencies of CD107a$^+$ cells among total splenic NK cells (CD3$^-$CD19$^-$NK1.1$^+$NKp46$^+$) and the indicated subpopulations thereof from control versus Raptor$^{\Delta NK}$ mice after coculture with Yac-1 targets in the presence of GolgiStop for 6 h. **e** The cumulative ratio of RMA-S cell rejection 18 h after intraperitoneal injection of a 1:1 mixture of RMA and RMA-S cells into each mouse. **f, g** Flow cytometric analysis and cumulative data depicting the expression of perforin and granzyme B (**f**), CD98 and CD71, and uptake of 2-NBDG (**g**) within total splenic NK cells (CD3$^-$CD19$^-$NK1.1$^+$NKp46$^+$) and the indicated subpopulations thereof. The MFI was calculated relative to total NK cells or to the CD27SP subset of NK cells from control littermates. CD27SP, DP, and CD11bSP represent CD27$^+$CD11b$^-$, CD27$^+$CD11b$^+$, and CD27$^{--}$CD11b$^+$ NK cell subsets, respectively. Each dot represents one mouse, and the experiments were replicated 3 (**c**), 4 (**g**), 5 (**a**, **b**), 7 (**d**), or 9 (**f**) times; error bars represent SD; *$p < 0.05$, **$p < 0.01$, ***$p < 0.001$, and ****$p < 0.0001$; unpaired two-tailed Student's $t$ test with Welsh's correction (**a–d**, **f**, **g**); generalized linear models (**e**). Each of the shapes represents a matched pair of littermates in an individual experiment, and the horizontal red line represents the median (**e**)

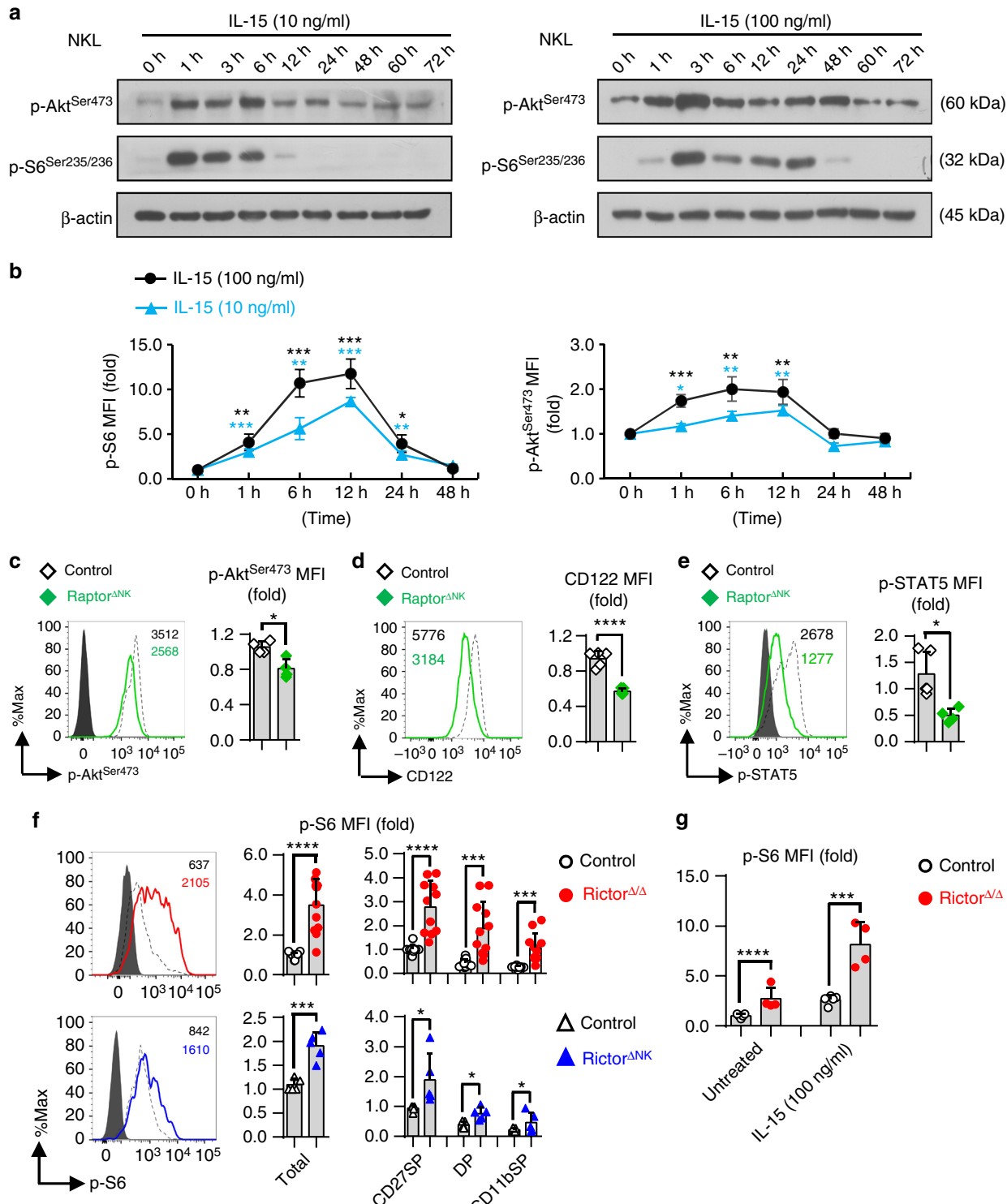

in the relative proportion and absolute quantity of NK cells due to Rictor deficiency (Fig. 5b). Further analysis of NK cell maturation revealed that compared with the Rictor$^{\Delta NK}$ mice, the Rictor$^{\Delta NK}$Raptor$^{fl/+}$ mice possessed a significantly higher proportion of immature CD27$^+$CD11b$^-$ NK cells and a reduced proportion of mature CD27$^-$CD11b$^+$ NK cells, concomitant with fewer CD43$^+$KLRG$^+$ mature NK cells (Fig. 5c, d).

We further compared the expression profiles of Tbx21 and Eomes, both of which are T-box family transcription factors with critical roles in orchestrating NK cell development[33–35], in each

subset of NK cell from the Rictor$^{\Delta/\Delta}$, Raptor$^{\Delta NK}$ and Rictor$^{\Delta NK}$-Raptor$^{fl/+}$ mice. The data revealed that *Tbx21* mRNA and protein expression levels were decreased by approximately 50% in total and in each subset of NK cells from the Rictor$^{\Delta/\Delta}$ mice compared with the levels observed in the control mice (Fig. 5e). Our research findings are in accordance with a previous study showing that mTORC2 positively regulates Tbx21 expression via p-Akt$^{Ser473}$ in Th1 cells[36]. In contrast, *Eomes* mRNA and protein expression levels were relatively unaffected in NK cells from Rictor$^{\Delta/\Delta}$ mice (Fig. 5e). However, in Raptor$^{\Delta NK}$ mice, the

**Fig. 4** Crosstalk between mTORC1 and mTORC2 during NK cell activation. **a** Immunoblotting to detect the phosphorylation (p-) of S6 at ser235/236 (p-S6$^{ser235/236}$) and Akt at Ser473 (p-Akt$^{ser473}$) in NKL cells after stimulation with IL-15 (10 ng/ml) (left) or IL-15 (100 ng/ml) (right) for the indicated time periods. β-Actin was used as an internal control. The data shown are representative of three independent experiments. Uncropped gels are shown in Supplementary Fig. 8. **b** Intracellular flow cytometric analysis of p-S6$^{ser235/236}$ and p-Akt$^{ser473}$ in murine splenic NK cells (CD3$^-$CD19$^-$NK1.1$^+$) after stimulation with IL-15 (100 or 10 ng/ml) for the indicated time periods. The MFI of p-S6$^{ser235/236}$ or p-Akt$^{ser473}$ at each time point was normalized to the baseline reading at the 0-h time point for each individual mouse. Four independent individual mice were used. **c–e** Flow cytometric analysis and cumulative results for p-Akt$^{Ser473}$ (**c**), CD122 (**d**), and p-STAT5$^{Y694}$ (**e**) in splenic NK cells (CD3$^-$CD19$^-$NK1.1$^+$) from control versus Raptor$^{ΔNK}$ mice. **f** Intracellular flow cytometric analysis and cumulative results for p-S6$^{ser235/236}$ in total splenic NK cells (CD3$^-$CD19$^-$NK1.1$^+$) and the indicated subpopulations thereof from control versus Rictor$^{Δ/Δ}$ (top) or control versus Rictor$^{ΔNK}$ (bottom) mice. **g** Statistical quantification of p-S6$^{ser235/236}$ (MFI relative to the untreated control littermates) in splenic NK cells (CD3$^-$CD19$^-$NK1.1$^+$) with or without IL-15 stimulation for 4 h ex vivo. The dashed line indicates the control group, and the solid line indicates the gene knockout group. The MFI was calculated relative to total NK cells or to the CD27SP subset of NK cells from control littermates. CD27SP, DP, and CD11bSP represent CD27$^+$CD11b$^-$, CD27$^+$CD11b$^+$, and CD27$^-$CD11b$^+$ NK cell subsets, respectively. Each dot represents one mouse, and all the experiments were replicated 3 (**c–e**, **f** bottom, **g**), or 7 (**f** top) times; error bars represent SD; *$p < 0.05$, **$p < 0.01$, ***$p < 0.001$, and ****$p < 0.0001$; one-way ANOVA (**b**, **g**); unpaired two-tailed Student's $t$ test with Welsh's correction (**c–f**)

mRNA and protein expression levels of both *Tbx21* and *Eomes* were reduced in total and in each subset of NK cells (Fig. 5f). Unexpectedly, the recovery of mTORC1 activity in NK cells from the Rictor$^{ΔNK}$Raptor$^{fl/+}$ mice resulted in levels of Tbx21 and Eomes protein expression comparable with those in NK cells from the Rictor$^{ΔNK}$ mice (Fig. 5g).

Taken together, these results suggest that mTORC2 and mTORC1 may control NK cell homeostasis and maturation in a cooperative and nonredundant manner by controlling the expression of Tbx21 and Eomes in a cell-intrinsic manner.

**mTORC2 blocks NK cell effector function by inhibiting mTORC1.** Regarding NK cell effector function, the data showed that normalization of mTORC1 activity in the Rictor$^{ΔNK}$Raptor$^{fl/+}$ mice reversed the enhanced NK cell effector functions that occurred during Rictor deficiency, such as higher basal expression levels of perforin, granzyme B, and CD71 and increased glucose uptake, but not the elevated expression of CD107a, in total as well as in each subset of NK cells (Fig. 6a–c). This finding indicates that enhanced per-cell NK cell cytolytic functions are dependent on mTORC1 activity.

To further exclude the possibility that the reduced NK cell effector functions in the Rictor$^{ΔNK}$Raptor$^{fl/+}$ mice were caused by a greater number of immature NK cells compared with those in the Rictor$^{ΔNK}$ mice, we further treated splenic NK cells from the Rictor$^{Δ/Δ}$ mice with rapamycin to attenuate mTORC1 activity, followed by the analysis of NK cell functional markers. An initial study showed that 0.1 nM rapamycin was sufficient to reverse the enhanced mTORC1 activity in Rictor-deficient NK cells to levels comparable with those detected in control NK cells (Fig. 6d). Consistently, the enhancement of several functional markers, such as perforin, granzyme B and 2-NBDG, in NK cells from the Rictor$^{Δ/Δ}$ mice were also reduced by 0.1 nM rapamycin treatment to such an extent that there was no significant difference in the expression of these markers between NK cells treated with rapamycin from the Rictor$^{Δ/Δ}$ mice and NK cells from the control mice (Fig. 6e). However, 0.1 nM rapamycin treatment did not affect levels of CD71, CD98, and CD107a expression in splenic NK cells from the Rictor$^{Δ/Δ}$ mice (Fig. 6f, g), which was consistent with our observations in the Rictor$^{ΔNK}$Raptor$^{fl/+}$ mice.

Overall, the above findings demonstrate strong evidence to support that mTORC2 is dependent on its inhibition of mTORC1 activity for its capacity to constrain NK cell effector functions in a cell-intrinsic manner, although the influence of other factors could not be excluded.

**Mechanisms of mTORC2 repressing mTORC1 activity in NK cells.** To explore the mechanisms by which mTORC2 regulates

mTORC1 activity, we first measured the level of classical Akt signaling activation required for mTORC1 activation upon acquiring extracellular or intracellular signaling[8]. Unexpectedly, the levels of Akt phosphorylation at Thr308 were unaltered in Rictor-deficient NK cells (Fig. 7a). In addition, a specific Akt inhibitor, AKTi-1/2[37], only slightly affected mTORC1 activity, without statistically difference, in Rictor-deficient NK cells, although it obviously inhibited mTORC1 activity in WT NK cells (Fig. 7b). These results suggest that classical Akt signaling is not responsible for the enhanced mTORC1 activation in Rictor-deficient NK cells, and other signaling pathways[8] might compensate for the phosphorylation of these substrates.

Because STAT5 signaling is a critical downstream pathway of IL-15 activation[32] and our finding that Rictor-deficient NK cells showed a robust response to ex vivo IL-15 stimulation (Fig. 4g), we next analyzed the level of CD122 and p-STAT5 in Rictor-deficient NK cells. Although expression of the IL-15 receptor CD122 was normal on the surface of NK cells from the Rictor$^{Δ/Δ}$ mice (Fig. 7c), the level of p-STAT5 was increased in a pattern similar to that of p-S6$^{Ser235/236}$ with or without IL-15 stimulation (Fig. 7d). To further explore the role of enhanced STAT5 signaling in mTORC2-mediated regulation of mTORC1 activity, we used the STAT5 inhibitor STAT5-IN[38] to treat Rictor-deficient NK cells. Interestingly, the data revealed that STAT5 inhibition could reverse the enhanced mTORC1 activity (Fig. 7e). As such, our data revealed the involvement of STAT5 signaling in mTORC1 activity regulation, showing that STAT5 participates in the regulation of mTORC1 activity by mTORC2. The STAT5 signaling pathway has been reported to positively regulate the cytotoxicity of NK cells[39], and treatment of NK cells with STAT5-IN not only reduced perforin and granzyme B expression in Rictor-deficient NK cells, consistent with results following the treatment of these cells with 0.1 nm rapamycin (Fig. 7f), but also partially inhibited the expression of CD107a (Fig. 7g). These data indicated that mTORC2 possesses roles in repressing mTORC1 activity by regulating STAT5 signaling in NK cells.

SLC7A5 is the only expressed light chain subunit of the heterodimeric system ʟ-amino acid transporters and is robust increased after IL-2/IL-12 stimulation in NK cells, which is responsible for mTORC1 activation during NK cell activation[37]. Recent studies reveal that STAT5 is important for the expression of SLC7A5 in human primary NK cells[40], which prompted us to test whether SLC7A5 was involved in the enhanced STAT5-mTORC1 signaling in Rictor-deficient NK cells. We firstly successfully validated the reliability of SLC7A5 antibodies for flow cytometric analysis following the same conditions used in the study of Loftus et al.[37] (Supplementary Fig. 4). Then, we found SLC7A5 was obviously increased on total NK cells and the indicated NK subsets in Rictor$^{Δ/Δ}$ mice (Fig. 7h), which could be

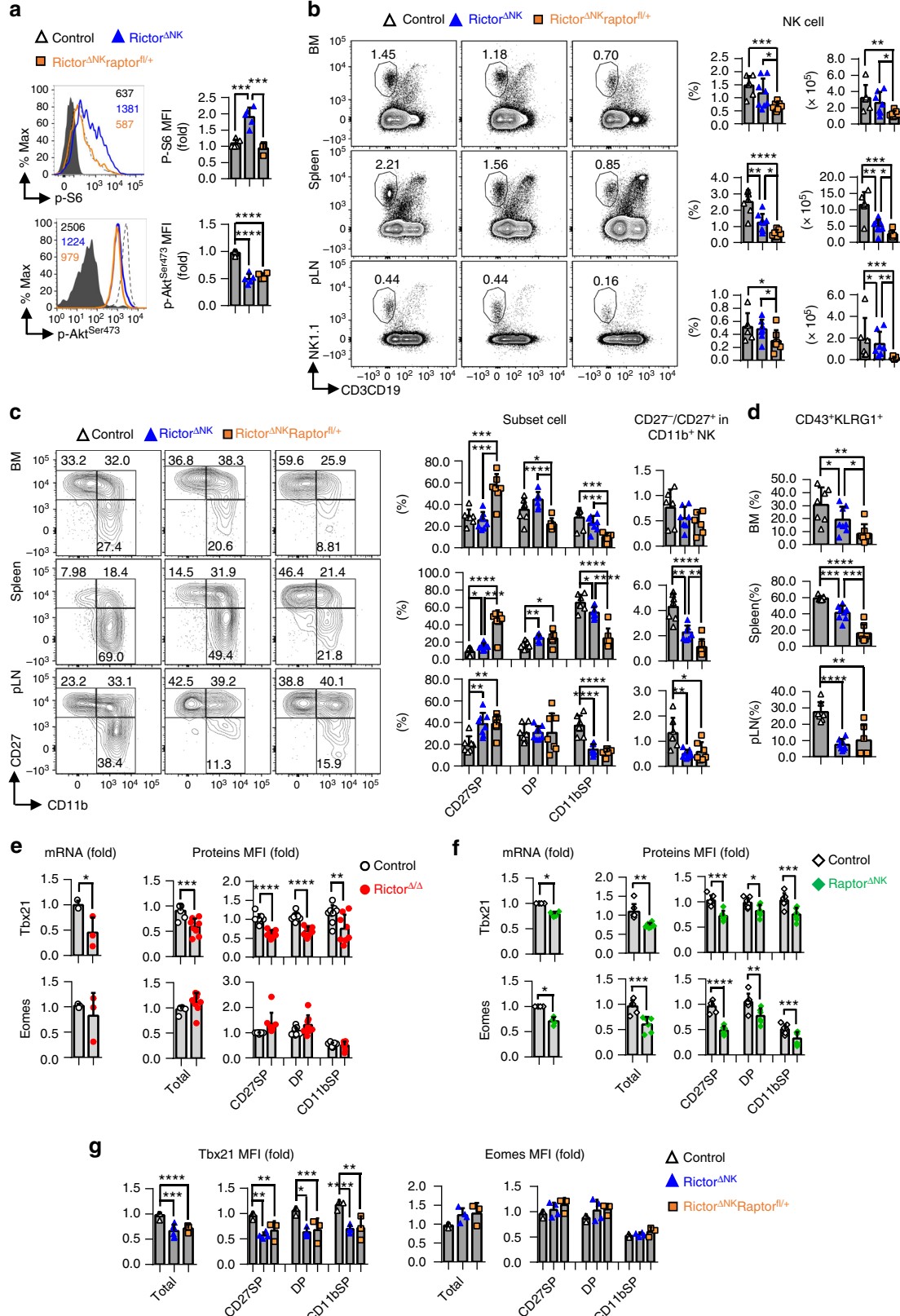

reversed by STAT5-IN treatment (Fig. 7i). Furthermore, a specific inhibitor of SLC7A5, 2-amino-2-norbornanecarboxylic acid (BCH)[37], could also inhibit the enhanced mTORC1 activity in Rictor-deficient NK cells (Fig. 7j). Overall, our results indicated that the increased mTORC1 activity in Rictor-deficient NK cells was mainly attributed to the enhanced STAT5–SLC7A5 axis.

**Tbx21[+/−] mice show normal NK cell effector function.** Finally, there remained an interesting question concerning the enhanced effector functions we observed in NK cells from the Rictor[Δ/Δ] mice, which seems contrary to the reduction in Tbx21 expression, especially given that previous studies have reported reduced NK cell effector functions in mice with homozygous *Tbx21* knockout

**Fig. 5** mTORC2 and mTORC1 promote NK cell maturation by controlling the expression of Tbx21 and Eomes in a cooperative and nonredundant manner. **a** Intracellular flow cytometric analysis and cumulative results for the phosphorylation (p-) of S6 at ser235/236 (p-S6$^{ser235/236}$) (top) and Akt at Ser473 (p-Akt$^{Ser473}$) (bottom) in splenic NK cells (CD3$^-$CD19$^-$NK1.1$^+$) from mice of the indicated genotype. **b** Cumulative ratio and enumeration of NK cells (CD3$^-$CD19$^-$NK1.1$^+$) in the BM, spleen, and peripheral lymph nodes (pLNs) from mice of the indicated genotype. **c** Flow cytometric analysis and cumulative frequencies of subpopulations of NK cells (CD3$^-$CD19$^-$NK1.1$^+$NKp46$^+$) in the BM, spleen, and pLNs (left) and the calculated ratio of CD27$^-$ versus CD27$^+$ cells among CD11b$^+$ NK cells (right). **d** The cumulative frequencies depicting the CD43$^+$KLRG1$^+$ subset of NK cells (CD3$^-$CD19$^-$NK1.1$^+$NKp46$^+$) in the BM, spleen, and pLNs from mice of the indicated genotype. **e, f** Tbx21 and Eomes mRNA and protein expression in control *versus* Rictor$^{\Delta/\Delta}$ mice (**e**) and control versus Raptor$^{\Delta NK}$ mice (**f**), as assessed by quantitative RT-PCR and flow cytometry, respectively. Purified splenic NK cells were used for quantitative RT-PCR (left). Cumulative data for Tbx21 and Eomes protein expression in total splenic NK cells (CD3$^-$CD19$^-$NK1.1$^+$NKp46$^+$) and the indicated subpopulations thereof were analyzed by flow cytometry (right). **g** Cumulative data for Tbx21 and Eomes protein expression in total splenic NK cells (CD3$^-$CD19$^-$NK1.1$^+$NKp46$^+$) and the indicated subpopulations thereof from control, Rictor$^{\Delta NK}$ and Rictor$^{\Delta NK}$Raptor$^{fl/+}$ mice. The control represents *Rictor*$^{fl/+}$/*Ncr1*-Cre$^+$ mice; Rictor$^{\Delta NK}$ represents *Rictor*$^{fl/fl}$/*Ncr1*-Cre$^+$ mice; and Rictor$^{\Delta NK}$Raptor$^{fl/+}$ represents *Rictor*$^{fl/fl}$/*Rptor*$^{fl/+}$/*Ncr1*-Cre$^+$ mice. The dashed line indicates the control group, and the solid line indicates the gene knockout group. The MFI was calculated relative to total NK cells or to the CD27SP subset of NK cells from control littermates. The subpopulations of NK cells are distinguished by surface expression of CD27 and CD11b, and CD27SP, DP, and CD11bSP represent the CD27$^+$CD11b$^-$, CD27$^+$CD11b$^+$, and CD27$^-$CD11b$^+$ NK cell subsets, respectively. Each dot represents one mouse, and all experiments were replicated 3 (**a, e** left, **f, g**), 4 (**e** right), or 5 (**b–d**) times; error bars represent SD; *$p < 0.05$, **$p < 0.01$, ***$p < 0.001$, and ****$p < 0.0001$; one-way ANOVA (**a–d, g**), unpaired two-tailed Student's *t* test (**e, f**)

---

($Tbx21^{-/-}$)[35, 41]. One plausible explanation for this discrepancy is that the reduction in Tbx21 expression, as observed in Rictor-deficient NK cells, may impair NK cell maturation while only slightly affecting the effector function. Consequently, we crossed $Tbx21$ heterozygous knockout ($Tbx21^{+/-}$) mice with congenic C57/B6L WT mice to generate littermate WT and $Tbx21^{+/-}$ mice. The data revealed that the $Tbx21^{+/-}$ mice nearly duplicated the phenotype of impaired NK cell homeostasis and maturation observed in the Rictor$^{\Delta NK}$ mice, including the ratio and quantity of NK cells and the profiles of CD11b and CD27 expression in NK cells (Supplementary Fig. 5a, b). Interestingly, in the $Tbx21^{+/-}$ mice, there was no appreciable influence on NK cell effector functions, including IFN-γ production following stimulation with PMA and ionomycin, CD107a expression following coculture with Yac-1 targets, and the basal expression levels of perforin or granzyme B (Supplementary Fig. 5c–e). In addition, several markers indicative of cell metabolic status, such as CD71 and CD98, as well as glucose uptake, were also unaltered in the $Tbx21^{+/-}$ mice (Supplementary Fig. 5f). Taken together, our results suggest that a single allele of $Tbx21$ in the $Tbx21^{+/-}$ mice is sufficient to maintain its influence on NK cell effector function but not on maturation.

## Discussion

Deciphering the specific functions of the mTOR signaling pathway in lymphocyte populations is of great importance for advanced understanding about and perhaps preventing or treating human diseases[8]. In comparison with the wealth of available data regarding mTORC1, information about the physiological and cellular functions of mTORC2 is relatively scarce[9]. In the current study, we identified the critical roles of mTORC1 and mTORC2 in regulating NK cell development and effector functions. NK cell-specific deletion of Raptor, the core protein of mTORC1, led to reduced NK cell maturation and effector functions. Genetic deletion of the core protein of mTORC2, Rictor, in NK cells during both the early and late stages of NK cell development resulted in profoundly impaired NK cell specification and maturation. Unexpectedly, Rictor-deficient NK cells displayed enhanced effector function via enhanced mTORC1 activity. Therefore, our results revealed for the first time a paradoxical role of mTORC2 in regulating NK cell development and effector functions. Notably, as Raptor or Rictor proteins were not completely deleted from NK cells in the Ncr1-Cre mice models, strictly speaking, the phenotypes we observed actually resulted from reduced expression but not complete loss of the Raptor or Rictor protein. However, neither Rictor nor Raptor loss would

affect the viability of any NK cell subsets (Supplementary Fig. 6), so we believe that the phenotype observed in our studies is indeed from NK cells lacking the proteins rather than just from the surviving subsets.

At present, the role of Tbx21 as a positive deterministic signal for late-stage NK cell maturation and effector function is widely accepted[35, 42]. We previously identified a transcription factor, Foxo1, negatively regulates Tbx21 transcription in NK cells[43]. However, the identification of the signaling pathway upstream of Tbx21 during NK cell development has remained a fundamental question. A previous study found that mTORC2 and mTORC1 are activated at a relatively higher level in immature NK cells and that their activity decreases in a coordinated manner as NK cell maturation progresses[12]. Deletion of $mTOR$ attenuates Tbx21 expression[12], while heightened mTORC1 activity (achieved via $Tsc1$ deletion) increases Tbx21 expression[13]. In the current study, we found that impaired mTORC2 activity via $Rictor$ deletion or diminished mTORC1 activity via $Rptor$ deletion reduced expression of Tbx21 mRNA and protein. We also found that mTORC2 regulated Tbx21 expression in a manner independent of mTORC1 activity. Taken together, our data supports the conclusion that important signaling regulators upstream of Tbx21 expression include mTORC2 and mTORC1, which integrate both extracellular and intracellular signals for controlling late-stage NK cell maturation.

The cellular and molecular functions of mTORC1 and mTORC2 occur in a highly population- and context-specific manner. Discerning the specific roles attributed to each of these two complexes in the context of cellular function is not only of great interest but also represents a formidable challenge[8]. We recently found that mTORC1 but not mTORC2 is essential for follicular regulatory T cell differentiation[44]. In the current study, we utilized both $Vav1$-Cre and $Ncr1$-Cre mice to delete $Rictor$ in both early- and late-stage NK cells. Using these population-specific approaches to delete $Rictor$ in NK developmental subsets, we found that mTORC2 activity promoted NK cell specification and homing to the spleen and downregulated CD27 while promoting terminal maturation of CD11b$^+$ NK cells. In contrast, mTORC1 activity constrained NK cell proliferation and survival in the BM, controlled NK cell homing to lymph nodes, and promoted the acquisition of CD11b and terminal maturation of CD11b$^+$ NK cells. Moreover, Rictor$^{\Delta NK}$ Rptor$^{fl/+}$ mice showed more severely impaired NK cell homeostasis and maturation than Rictor$^{\Delta NK}$ or Raptor$^{\Delta NK}$ mice. Taken together, our findings indicate that both mTORC2 and mTORC1 control NK cell homeostasis and maturation but in different ways and with some overlap.

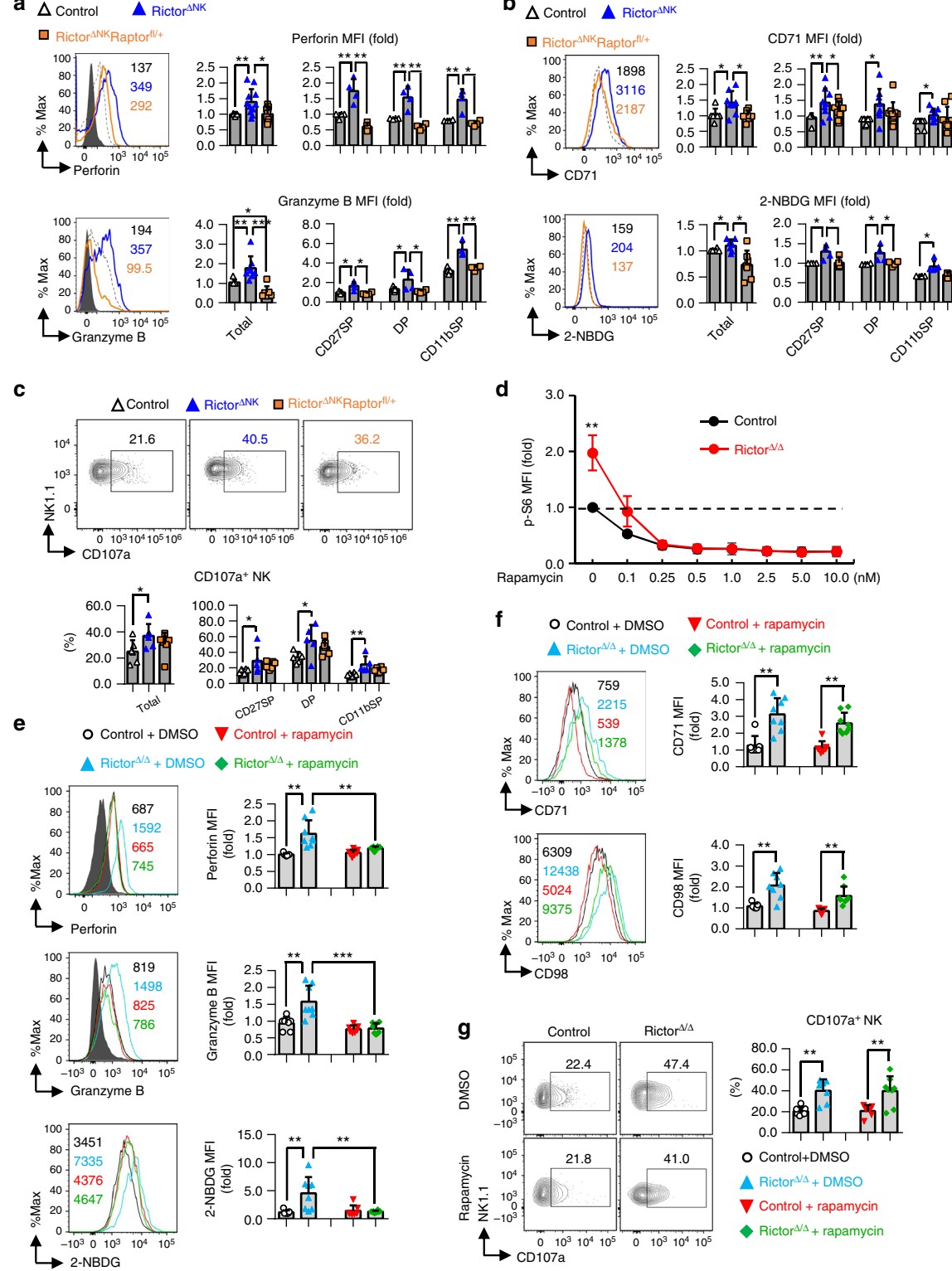

mTORC1 and mTORC2 are related yet distinct from each other in the context of various cellular and molecular processes[8]. For example, mTORC1 deficiency in both $CD4^+$ and $CD8^+$ T cells promotes mTORC2 activation, while mTORC2 deficiency does not influence mTORC1 activity in $CD4^+$ T cells[45], but it does slightly improve mTORC1 activity in $CD8^+$ T cells[46]. The means by which mTORC1 and mTORC2 cross-regulate each other's activity in the context of cellular function remains enigmatic. Here, we unraveled the specific regulatory circuits underlying the interplay between mTORC1 and mTORC2 in NK cells. In particular, mTORC1 positively regulates mTORC2 activity via IL-15 signaling by maintaining the expression of the IL-15 receptor CD122; in contrast, mTORC2 acts as a negative regulator of mTORC1 in controlling NK cell effector functions, and this regulation mainly involves inhibition on STAT5–SLC7A5 axis. Several lines of evidence support this conclusion: (i) the levels of mTORC2 and mTORC1 activity were coordinated in a similar fashion following cytokine activation;

**Fig. 6** mTORC2 restrains NK cell effector functions by inhibiting mTORC1 activity. **a, b** The expression of perforin and granzyme B (**a**), CD71, and uptake of 2-NBDG (**b**) in total splenic NK cells (CD3$^-$CD19$^-$NK1.1$^+$NKp46$^+$) and NK cell subsets from mice of the indicated genotypes. **c** Flow cytometric analysis of CD107a$^+$ cells among total splenic NK cells (CD3$^-$CD19$^-$NK1.1$^+$NKp46$^+$) or the NK cell subsets specified from mice of the indicated genotypes after coculture with YAC-1 targets. **d** The expression of phosphorylated (p-) S6 at ser235/236 (p-S6$^{ser235/236}$) in splenic NK cells (CD3$^-$CD19$^-$NK1.1$^+$) from control versus Rictor$^{\Delta/\Delta}$ mice after stimulation with rapamycin of the indicated concentrations for 6 h. The MFI of p-S6$^{ser235/236}$ at each concentration was normalized to the baseline reading (i.e., 0 nm rapamycin-treated control NK cells in a littermate pair). The horizontal dashed line indicates the normalized baseline reading. **e–f** The expression of perforin, granzyme B, uptake of 2-NBDG (**e**), and CD71 and CD98 (**f**) (MFI relative to the littermates of the control + DMSO group) in splenic NK cells from control versus Rictor$^{\Delta/\Delta}$ mice treated with DMSO (set as the solvent control) or 0.1 nM rapamycin for 6 h ex vivo. **g** Flow cytometric analysis of CD107a$^+$ splenic NK cells from control versus Rictor$^{\Delta/\Delta}$ mice following treatment with DMSO or 0.1 nM rapamycin, described in detail in the ONLINE METHODS. The control represents Rictor$^{fl/+}$/Ncr1-Cre$^+$ mice; Rictor$^{\Delta NK}$ represents Rictor$^{fl/fl}$/Ncr1-Cre$^+$ mice; and Rictor$^{\Delta NK}$Raptor$^{fl/+}$ represents Rictor$^{fl/fl}$/Rptor$^{fl/+}$/Ncr1-Cre$^+$ mice. The dashed line indicates the control group, and the solid line indicates the gene knockout group. The MFI was calculated relative to total NK cells or to the CD27SP subset of NK cells from control littermates. The subpopulations of NK cells are distinguished by surface expression of CD27 and CD11b, and CD27SP, DP, and CD11bSP represent the CD27$^+$CD11b$^-$, CD27$^+$CD11b$^+$, and CD27$^-$CD11b$^+$ NK cell subsets, respectively. Each dot represents one mouse, and all experiments were replicated 3 (NK subsets in **a**), 4 (**c, e–g**), 5 (**b**), or 7 (total NK cells in **a**) times; error bars represent SD; *$p < 0.05$, **$p < 0.01$, ***$p < 0.001$, and ****$p < 0.0001$; one-way ANOVA (**a–c, e–g**). Control, $n = 3$; Rictor$^{\Delta/\Delta}$, $n = 5$ in 3 independent experiments (**d**)

namely, the activation of both mTORC2 and mTORC1 was robustly increased at an early time point and then was gradually downregulated. However, disruption of mTORC2 activity via *Rictor* deficiency led to a twofold increase in mTORC1 activity in NK cells. Impairment of mTORC1 via deletion of *Rptor* resulted in downregulation of mTORC2 activity due to reduced responsiveness to IL-15 caused by impaired CD122 expression, which are downstream effects of mTORC1 deficiency[12]. In conjunction with this phenomenon, the function of several cellular processes, such as nutrient uptake and mitochondrial activity, known to occur downstream of mTORC1 signaling was elevated in Rictor-deficient NK cells on a "per-cell" basis[47]. (ii) Restoring the elevated levels of mTORC1 activity seen in Rictor-deficient NK cells to levels observed in WT NK cells (achieved either by genetically deleting half the copies of *Rptor* or treatment with a low concentration of rapamycin) attenuated the enhanced NK cell effector functions caused by mTORC1 hyperactivation. (iii) Rictor-deficient NK cells also showed increased STAT5 activity and SLC7A5 expression. Although we could not definitely exclude the contribution of the artifact of Rictor deletion in participating in the enhanced mTORC1 activity, the inhibition of STAT5 or SLC7A5 activity by inhibitors was able to reverse the elevation of mTORC1 activity, to a similar level in WT NK cells, in Rictor-deficient NK cells. Furthermore, blocking SLC7A5 activity by a specific inhibitor, BCH, could also reverse the enhanced mTORC1 activity in Rictor-deficient NK cells. In addition, our data showed that the enhanced mTORC1 activity in Rictor-deficient NK cells was independent of AKT signaling. Together with previous finding that SLC7A5 regulates mTORC1 activity independent of AKT signaling[37], it further highlights the critical roles of STAT5–SLC7A5 axis in the interplay between mTORC1 and mTORC2 in NK cells.

We believe that this interplay between mTORC1 and mTORC2 will prove valuable in attempting to regulate the functions of NK cells. During NK cell activation, mTORC2 activity, mainly through STAT5–SLC7A5 axis, counteracts the effects of constitutive mTORC1 hyperactivation, which avoids activation-induced death of NK cells[13]. In some circumstances, such as NK exhaustion, once NK cell mTORC1 activity declines, the activity of mTORC2 decreases concomitantly, thus releasing the repression of the STAT5 signaling pathway and, in turn, rescuing mTORC1-mediated NK cell effector functions. These findings reveal that the intrinsic STAT5 signaling pathway not only acts as a passive responder to environmental signals, but also plays an active role in keeping NK cell homeostasis by regulating mTOR signaling. Given the evidence from other research laboratories

and our own indicating that a reduction in mTORC2 activity promotes CD8$^+$ memory T cell differentiation and response[46] as well as NK cell cytolytic function, the exploration of inhibitors that selectively repress mTORC2 activity would be of immense research benefit, especially considering that mTORC2 abnormalities are associated with many human cancers, including prostate, breast, and non–small-cell lung cancers, glioblastoma, and T cell acute lymphoblastic leukemia[8, 48].

Recently, Yang et al.[49] reported that mTORC1 and mTORC2 exerted divergent roles in regulating NK cell development, but both promoted NK cell effector function. The chief reason for the apparent discrepancy between the results of Yang et al. and the current study is the different controls for Rptor$^{\Delta NK}$ or Rictor$^{\Delta NK}$ mice. Yang et al. used *Rptor$^{fl/fl}$* and *Rictor$^{fl/fl}$* mice, while we used *Rptor$^{fl/+}$/Ncr1-Cre$^+$* and *Rictor$^{fl/+}$/Ncr1-Cre$^+$*, as the controls for Rptor$^{\Delta NK}$ or Rictor$^{\Delta NK}$ mice, respectively. Several published literatures have clearly demonstrated that NKp46, encoded by *Ncr1*, is an activating receptor for NK cells, which plays critical roles in repressing the growth and metastasis of tumors, especially melanoma, via IFN-γ production[50]. Previous studies have demonstrated that despite *Ncr1*-Cre$^+$ heterozygotes mouse showed no effect on the percentage of NKp46$^+$ NK cells, a significant downregulation in the density of cell surface NKp46 is found in this line[51]. Therefore, several literatures used *Ncr1*-Cre$^+$ X$^{fl/WT}$ (X indicates target gene) as the control in their studies[43, 52–54]. By using the appropriate controls in the current study, we uncovered the essential characteristic of mTORC2 in regulating NK cell effector function, the first claimed factor oppositely regulating NK cell maturation and effector function.

In conclusion, we have revealed that mTORC1 and mTORC2 regulate NK cell development and maturation in a unique but nonredundant manner by differentially regulating Tbx21 and Eomes expression. mTORC1 sustains mTORC2 activity by maintaining CD122-mediated IL-15 signaling, whereas mTORC2 acts as a repressor of mTORC1 to control NK cell effector functions mainly through restraining STAT5–SLC7A5 axis. Our findings broaden the current knowledge regarding the interplay that occurs among mTOR and STAT5 signaling in NK cells and have the potential to provide a new strategy for targeting NK cells for antitumor therapy and for potentially improving NK-mediated viral clearance by repressing mTORC2 activation.

## Methods
**Mice and cell lines**. All mice were bred and housed in pathogen-free conditions, which were kept at 22–25 °C, and had 12-h light/dark cycle, periodic air changes, and free access to water and food, in the Experimental Animal Center of the Third Military Medical University (Chongqing, China). All animal experiments

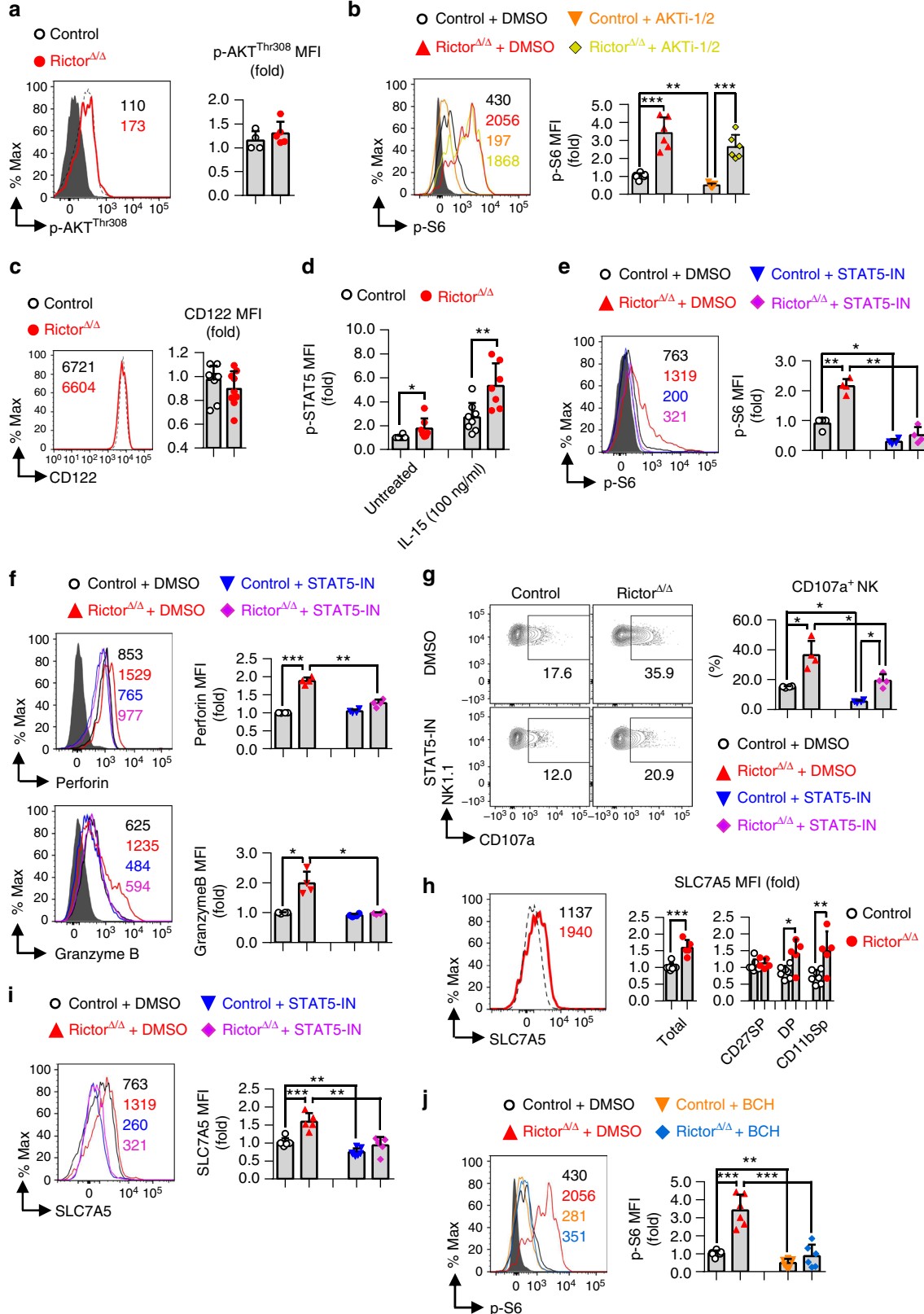

performed in this study were approved by the Animal Ethics Committee of the Third Military Medical University, following the guidelines of the Institutional Animal Care and Use Committees of the Third Military Medical University.

C57BL/6 *Vav1*-Cre (B6.Cg-Tg(*Vav1*-cre)A2Kio/J), *Rictor*fl/fl (B6.Cg-*Rictor*tm1.1Klg/Sjm J), *Rptor*fl/fl (B6.Cg-*Rptor*tm1.1Dmsa/J), Tbx21$^{-/-}$ (B6.129S6-*Tbx21*tm1Glm/J) mice were purchased from The Jackson Laboratory. C57BL/6

*Ncr1*-Cre (C57BL/6-*Ncr1*tm1(iCre)/Bcgen) mice were a kind gift from Beijing Biocytogen (Beijing, China). All mice were sacrificed at 8–12 weeks for experiments and littermates were used as controls for all the analyses.

YAC-1 and NKL were purchased commercially with authentication from the American Type Culture Collection and Biomedical Analysis Center of Army Medical University, China, respectively. RMA and RMA-S cells were kind gifts from Prof. André´ Veillette, which have previously been used in his studies[55]. All

**Fig. 7** Mechanisms of mTORC2 repression of mTORC1 activity in NK cells. **a** Intracellular flow cytometric analysis depicting the phosphorylation (p-) of Akt at Thr308 (p-Akt$^{Thr308}$) in total splenic NK cells (CD3$^-$CD19$^-$NK1.1$^+$). **b** The expression of p-S6$^{ser235/236}$ in splenic NK cells from control versus Rictor$^{\Delta/\Delta}$ mice after treatment with DMSO (set as the solvent control) or 2 μM AKTi-1/2 for 6 h ex vivo. **c** The abundance of CD122 expression on splenic NK cells (CD3$^-$CD19$^-$NK1.1$^+$NKp46$^+$) from control versus Rictor$^{\Delta/\Delta}$ mice, analyzed by flow cytometry. **d** Statistical quantification of p-STAT5$^{Y694}$ in splenic NK cells (CD3$^-$CD19$^{---}$NK1.1$^+$) with or without IL-15 stimulation for 4 h ex vivo. **e–g, i** Statistical quantification of p-S6$^{ser235/236}$ (**e**), perforin and granzyme B (**f**), frequencies of CD107a$^+$ NK cells (**g**), SLC7A5 expression (**i**) in splenic NK cells from control versus Rictor$^{\Delta/\Delta}$ mice after treatment with DMSO or 100 μM STAT5-IN for 6 h ex vivo. **h** Statistical quantification of SLC7A5 expression on both total NK cells and the indicated subsets. **j** The expression of p-S6$^{ser235/236}$ in splenic NK cells from control versus Rictor$^{\Delta/\Delta}$ mice after treatment with DMSO or 25 mM BCH for 6 h ex vivo. The MFI was relative to the littermates of control + DMSO group (**b, e, f, i, j**). Each dot represents one mouse, and all experiments were replicated 3 (**a, b, e–j**), 5 (**d**), 6 (**c**) times; error bars represent SD; *$p < 0.05$, **$p < 0.01$, ***$p < 0.001$; unpaired two-tailed Student's $t$ test (**a, c, h**); one-way ANOVA (**b, d–g, i, j**)

the cell lines had been tested and confirmed negative for mycoplasma contamination with a Mycoplasma PCR Detection Kit (Sigma-Aldrich). None of the cell lines used in this paper are listed in the database of commonly misidentified cell lines maintained by ICLAC.

**Antibodies and flow cytometry**. Antibodies used for flow cytometry were commercially purchased and are listed as followings: anti-CD3 (17A2, dilution 1/200, #100203, 100236, 100229, BioLegend), anti-CD19 (6D5, dilution 1/200, #115505, 115512, 115541, BioLegend), anti-NK1.1 (PK136, dilution 1/100, #108739, 108741, 108707, 108723, BioLegend), anti-NKp46 (29A1.4, dilution 1/100, #137604, 137619, 137607, 137612, BioLegend), anti-CD27 (LG.7F9, dilution 1/200, #25-0271-8, eBioscience), anti-CD11b (M1/70, dilution 1/200, #101227, BioLegend), anti-p-Akt$^{Ser473}$ (SDRNR, dilution 1/50, #17-9715-42, eBioscience), anti-p-S6$^{Ser235/236}$ (D57.2.2E, dilution 1/100, #5316, Cell Signaling Technology), anti-CD122 (TM-Beta1, dilution 1/100, #562960, BD Biosciences), anti-Ter119 (TER-119, dilution 1/100, #557915, BD Biosciences), anti-Gr-1 (RB6-8C5, dilution 1/100, #553127, BD Biosciences), anti-Ki-67 (B56, dilution 1/66, #558615, BD Biosciences), anti-CD43 (S7, dilution 1/200, #561857, BD Biosciences), anti-KLRG1 (2F1, dilution 1/200, #740156, BD Biosciences), anti-IFNγ (XMG1.2, dilution 1/300, #562019, BD Biosciences), anti-CD107a (1D4B, dilution 1/66, #121613, Bio-Legend), anti-Perforin (eBioOMAK-D, dilution 1/66, #12-9392-82, eBioscience), anti-Granzyme B (NGZB, dilution 1/66, #12-8898-82, eBioscience), anti-CD98 (RL388, dilution 1/66, #12-0981-81, eBioscience), anti-CD71 (C2, dilution 1/66, #553266, BD Biosciences), anti-p-STAT5$^{Y694}$ (47/Stat5(pY694), dilution 1/50, #612599, BD Biosciences), anti-T-bet (4B10, dilution 1/50, #12-5825-82, eBioscience), anti-Eomes (Dan11mag, dilution 1/66, #25-4875-82, eBioscience), anti-CD16/CD32(2.4G2, dilution 1/100, #553141, BD Biosciences), anti-Raptor (#514208, dilution 1/100, #IC5957G-100UG, R&D SYSTEMS), anti-Rictor (7B3, dilution 1/66, #NBP1-51645PE, NOVOUS BIOLOGICALS), anti-p-AKT$^{Thr308}$ (D25E6, dilution 1/50, #13842, Cell Signaling Technology), anti-SLC7A5 (poly-clonal, dilution 1/50, #bs-10125R-AF647, Bioss ANTIBODIES). We confirmed the species reactivity for every antibody according to the official directions and performed preliminary experiments to determine the appropriate dilution for all the antibodies.

Standard protocols were followed for flow cytometry[43]. Briefly, single-cell suspensions were obtained from the BM, spleen, and peripheral lymph nodes (pLN), and if necessary, cells were counted with Fuchs-Rosenthal Counting Chamber. For surface markers, antibodies were stained at room temperature and in the dark for 15 min in staining buffer (phosphate-bufferd saline (PBS) containing 2% mouse serum, 2% horse serum, and anti-CD16/CD32 blocking antibodies) and then washed with PBS.

For intracellular IFN-γ staining, cells were stimulated with PMA and ionomycin (eBioscience) plus BD Golgi Plug™ protein transport inhibitor (BD Biosciences) for 6 h, then cells were stained with Fixation/Permeabilization Solution Kit (BD Biosciences) following the manufacturer's instructions.

For staining of phosphorylated proteins, cells were fixed with IC Fixation Buffer (eBioscience) for 30–60 min at room temperature and then permeabilized with 1 ml −80 °C prechilled absolute methanol for at least 30 min at −80 °C. After a wash with staining buffer, intracellular phosphorylated proteins, and surface markers were stained together for 1 h at room temperature in the dark. All other intracellular proteins were stained according to manufacturer's instructions using Foxp3/Transcription Factor Staining Buffer Set Kit (eBioscience).

For apoptosis detection, an Annexin V Apoptosis Detection Kit with 7-AAD (BioLegend) was used following the manufacturer's instructions. Briefly, after staining with surface markers, cells were resuspended in 100 μl of Annexin V Binding Buffer containing 5 μl of Annexin V-APC and 5 μl of 7-AAD. Cells were then incubated in the dark for 15 min, and 400 μl of Annexin V Binding Buffer was added to each tube before flow cytometric analysis.

All flow cytometry was carried out on a BD FACSVerse™ or BD FACSCanto™, and data were analyzed with FlowJo software (Treestar). The gating or sorting strategies for all flow cytometry analysis in this study are included in Supplementary Fig. 7.

**NK cell cytotoxicity assay**. For CD107a expression analysis[43], splenic cells were mixed with Yac-1 cells at a 2:1 E:T ratio in a V-bottom 96-well plate in the presence of anti-CD107a antibody (BD Biosciences, 1D4B) and BD GolgiStop™ protein transport inhibitor (BD Biosciences). Cells were then centrifuged at 150$g$ for 3 min, cocultured for 6 h, and then collected for staining of additional surface markers and detection by flow cytometry.

For ex vivo NK cell killing assay, NK cell subsets were sorted based on the expression of CD27 and CD11b using a BD FACSAria™ III and RMA-S cells were labeled with 20 μM Calcein AM (BD Pharmingen) for 0.5 h at 37 °C. Then, 7500 labeled RMA-S cells were co-cultured with indicated NK subsets with an E:T ratio = 1:1 in a 96-well V-bottom plate for 4 h at 37 °C. Only target cells were set to measure the spontaneous release and a final concentration of 0.2% Triton X-100 lysis buffer was added to target cells to measure the maximal release. The fluorescence of the supernatant was measured with an excitation wavelength of 495 nm and an emission wavelength of 520 nm[37]. Three duplicate wells were set for every sample and the average was used for further analysis.

For the in vivo RMA-S clearance assay[56], NK-resistant RMA and NK-sensitive RMA-S cells were labeled with CellTrace™ Far Red (Invitrogen) and CellTrace™ Violet (Invitrogen), respectively, following the manufacturer's instructions. Then, a 1:1 mixture of labeled RMA and RMA-S cells (2 million mixed cells per mouse) was administered into the mice by intraperitoneal injection, while 2 million mixed cells were simultaneously cultured in complete RPMI-1640 medium in vitro. After 18 h, the mice were euthanized by cervical dislocation and cells in the peritoneal cavity were collected by repeated washing with PBS containing 2 μM EDTA. Concurrently, cells from the parallel in vitro control cultures were also collected. Surviving RMA-S cells were identified by flow cytometry, and the ratio (%) of RMA-S cell rejection was calculated using the following formula: 100 × (1 −[percentage of residual CellTrace™ Violet labeled population in total CellTrace™ Violet labeled and CellTrace™ Far Red labeled population in mouse peritoneal cavity/percentage of residual CellTrace™ Violet labeled population in total CellTrace™ Violet labeled and CellTrace™ Far Red labeled population in vitro]).

**Measurement of glucose uptake**. The fluorescent glucose analog 2-(N-(7-nitro-benz-2-oxa-1,3-diazol-4-yl)amino)-2-deoxyglucose (2-NBDG) (Invitrogen) was used as an agent to measure NK cell glucose uptake. Freshly isolated cells were resuspended with prewarmed (37 °C) RPMI-1640 medium (Life Technologies) in the presence of 100 μM 2-NBDG, and cultured at 37 °C for 10 min. Cells were then stained for surface markers (CD3, CD19, NKp46, NK1.1, CD11b, and CD27) before detection via flow cytometry.

**Measurement of mitochondria activity**. Splenic cells were resuspended with pre-warmed (37 °C) RPMI-1640 medium in the presence of 20 nM MitoTracker® Green FM (Invitrogen) or TMRE (Invitrogen), and then cultured at 37 °C for 30 min in the dark. Surface marker were then stained for detection via flow cytometry. For MitoSOX staining, instead of MitoTracker® Green FM or TMRE, 5 μM MitoSOX™ (Invitrogen) was used. Following culture at 37 °C for 10 min in the dark, cells were then stained for surface markers for assessment by flow cytometry.

**NK cell purification and quantitative RT-PCR analysis**. Freshly isolated sple-nocytes were stained with anti-CD3 (17A2) and anti-NK1.1 (PK136) antibodies, and NK cells were sorted to >98% purity using a BD FACSAria™ III.

Total RNA was extracted from sorted murine NK cells with a Total RNA Purification Micro Kit (Norgen Biotek Corp.) according to the manufacturer's instructions. Total RNA was then reverse-transcribed into cDNA with a Bestar™ qPCR RT Kit (DBI Bioscience). Real-time PCR reactions were carried out with Bestar® SYBRGreen qPCR master mix (DBI Bioscience) using an ABI Prism 7700 Sequence Detector[57]. Relative mRNA expression levels were calculated by normalizing the relative cycle threshold value to the control group after normalization to the internal control, HPRT1. Primer pairs used are as follows:
mouse Tbx21-Forward, 5′-CAACCAGCACCAGACAGAGA-3′;
mouse Tbx21-Reverse, 5′- ACAAACATCCTGTAATGGCTTG-3′;
mouse Eomes-Forward, 5′-CAACTACCATTCATCCCATCAG-3′;
mouse Eomes-Reverse, 5′-CAGATTCATAAGAACCGATGTC-3′;
mouse HPRT1-Forward, 5′-GCTGGTGAAAAGGACCTCT-3′;
mouse HPRT1-Reverse, 5′-CACAGGACTAGAACACCTGC-3′.

**Cell stimulation**. Before stimulation, NKL cells were starved in complete RPMI-1640 medium without IL-2 overnight. The cells were then stimulated continuously with 10 or 100 ng/ml recombinant human IL-15 (BioVision) until the indicated time periods were reached. Cells were then collected for immunoblotting analysis.

Freshly isolated splenic cells were allowed to rest in RPMI-1640 medium in an incubator for 1 h and were then stimulated with or without 100 ng/ml recombinant murine IL-15 (BioVision) for 4 h or with 10 or 100 ng/ml recombinant murine IL-15 (BioVision) continuously for the indicated time period. Cells were then fixed by directly mixing them (together with their culturing medium) with an equal volume of IC Fixation Buffer (eBioscience) and were subsequently stained for flow cytometry with antibodies directed against intracellular phosphorylated proteins and surface markers using the abovementioned method.

For rapamycin treatment, freshly isolated splenocytes were treated with DMSO (set as the solvent control) or rapamycin at the indicated concentration for 6 h in vitro. Phosphorylation of S6; the expression of perforin, granzyme B, CD71, and CD98; and 2-NBDG uptake were each analyzed according to the flow cytometric staining protocols indicated above. For analysis of CD107a, splenocytes were pretreated with DMSO or 0.1 nM rapamycin for 3 h, followed by coculture with Yac-1 targets at a 2:1 E:T ratio in the presence of DMSO or 0.1 nM rapamycin for an additional 6 h, at which time they were analyzed by flow cytometric staining as described above.

STAT5-IN (CAS no. 285986-31-4), AKTi-1/2 (CAS no. 612847-09-3) were purchased from Med Chem Express; BCH (CAS no. 20448-79-7) was purchased from Sigma-Aldrich. Preliminary experiments were performed to confirm 100 μM as the working concentration for STAT5-IN. Freshly isolated splenocytes were treated with DMSO (set as the solvent control), 100 μM STAT5-IN, 2 μM AKTi-1/2[37], or 25 mM BCH[37] for 6 h in vitro. Then, S6 phosphorylation, or perforin and granzyme B expression were analyzed according to the flow cytometric staining protocols indicated above. For analysis of CD107a, splenocytes were also pretreated with DMSO or 100 μM STAT5-IN for 3 h, followed by co-culture with Yac-1 targets at a 2:1 E:T ratio in the presence of DMSO or 100 μM STAT5-IN for an additional 6 h, at which time they were analyzed by flow cytometric staining as described above.

**Immunoblotting**. The stimulated NKL cells or sorted splenic NK cells were harvested and equal number of cells were directly lysed in 2× Laemmli buffer (Bio-Rad) supplemented with 2.5% β-mercaptoethanol and boiled for 6 min. Proteins were separated by 8–12% sodium dodecyl sulfate polyacrylamide gel electrophoresis and then transferred onto nitrocellulose membranes. Membranes were then blocked using 5% TBST-diluted skim milk for 1 h at room temperature, and incubated with primary antibodies at 4 ℃ for about 12–18 h. The primary antibodies used were as follows and diluted at 1/1000: Phospho-S6 Ribosomal Protein (Ser235/236) (D57.2.2E) Rabbit mAb (#4858, Cell Signaling Technology), Phospho-Akt (Ser473) (D9E) Rabbit mAb (#4060, Cell Signaling Technology), Rictor (53A2) Rabbit mAb (#2114, Cell signaling Technology), Raptor (24C12) Rabbit mAb (#2280, Cell signaling Technology). β-Actin (13E5) Rabbit mAb (#4970, Cell Signaling Technology) or GAPDH (14C10) Rabbit mAb (#2118, Cell signaling Technology) served as an internal control. Anti-rabbit HRP (ZSGB-BIO, ZB-2301) was used as a secondary antibody, followed by detection with Pierce™ ECL Western Blotting Substrate (Thermo Scientific™, 32209). Uncropped gels are shown in Supplementary Fig. 8.

**Statistical analysis**. When two independent groups were compared, if the data were normally distributed, an unpaired Student's *t* test was used for comparison, and if not, a nonparametric method, such as the Wilcoxon rank sum test, was used. When three or more independent groups were compared, one-way ANOVA was applied under the assumptions of normality and equal variance. Linear mixed models were used to analyze the correlation between repeated measures from the same mouse, while generalized linear models were used in the randomized block design with litters as the block factor. For multiple comparisons, *p* values were adjusted with Holm's procedure. All tests were two-sided, and a *p* value ≤ 0.05 was considered statistically significant.

It is generally considered that a sample size containing at least three replicates can provide adequate statistical power in biochemical analysis. Besides, we were sure that the mice could generate enough cells for analysis through preliminary studies. Systematic randomization and blinding was not used. No data captured was excluded from the subsequent analyses.

**Reporting Summary**. Further information on research design is available in the Nature Research Reporting Summary linked to this article.

## Data availability
The data that support the findings of this study are available from the authors on reasonable requests, see author contributions for specific data sets. A reporting summary for this article is available as a Supplementary Information file.

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

## Acknowledgments

We thank the core facility center of Third Military Medical University for cell sorting. We also thank Andre´ Veillette at Department of Medicine, University of Montre´al for providing the cell lines, RMA and RMA-S. This study was supported by grants from the National Natural Science Foundation of China (Nos. 81503083 and 81874313 to Y.-C.D.; Nos. 81473210, 81520108029, and 81773742 to X.L.; Nos. 81370631 and 81770207 to S. Y.; No. 81770216 to J.C.; U.S. NIH grant AI129582 to J.Y.).

## Author contributions

F.W. designed the research, performed experiments, analyzed the data and wrote the manuscript; M.M. designed the research, performed experiments and analyzed the data; B.M., Y.Y., P.H., W.L., X.P., Y.J., T.Y., H.L., X.G. and Y.-F.D. performed experiments; S. Y. and J.C. designed the research, performed the experiments and were involved in writing and reviewing the manuscript; M.N., T.H., L.Y. and J.Y. designed the research and were involved in writing and reviewing the manuscript; X.L. designed the research and supervised the study; Y.-C.D. devised the concept, designed the research, supervised the study, and wrote the paper.

## Additional information

**Competing interests:** The authors declare no competing interests.

