## [Peer Review file · Nature Communications]

Editorial Note: Parts of this peer review file have been redacted as indicated to remove third-party material where no permission to publish could be obtained.

Reviewers' comments:

Reviewer #1 (NK licensing, NKR)(Remarks to the Author):

The role of the mTOR pathway in cellular metabolism and immune cell homeostasis is well appreciated. In their manuscript, Wang et al use NK cell-specific deletion of Raptor and Rictor proteins to describe the interplay between mTORC1 and mTORC2, and their role in NK cell maturation and cytotoxic function. They demonstrate that both mTORC1 and mTORC2 are required for NK cell differentiation and maturation. However, they had opposing impact on NK cell function—mTORC1 promoted while mTORC2 inhibited NK cell cytotoxicity. mTORC2 was shown to suppress mTORC1 activity, which in turn regulates mTORC2 activity through IL-15 receptor (CD122) signalling. STAT5 was shown to be involved in mTORC2 suppression of mTORC1 activity.

The manuscript is well written and technically sound. Although many of the broad conclusions are not novel, the authors' meticulous data analysis and dissection of the underlying mechanisms behind mTORC effects on NK activity mean that this manuscript is of value to the scientific community. A major caveat in this study is the extent of NK cell-specific Raptor and Rictor deletion. The authors show deletion by flow cytometry; however, it seems like the two proteins are not completely lost in their respective deletion strains. As well, NK cell phenotypes shown in this manuscript are partial, which could possibly be due to incomplete loss of mTORC1 and 2 activity as assessed by S6 and Akt phosphorylation in these mice. The authors must explicitly highlight the possibility that their various models may not be complete knockouts, and discuss what impact this could have on the interpretation of their data.

Reviewer #2 (Immunometabolism, mTOR/AMPK)(Remarks to the Author):

While the authors have addressed some of my concerns satisfactorily, there are still some issues remaining.

Original comment:

Reviewer #2-1a. The cytotoxicity of NK cells towards tumour target cells changes during maturation with CD11b+CD27+ having the greatest killing potential (PMID: 12006976, PMID: 16424180, PMID: 26851218). Therefore, the increased in vitro cytotoxicity and in vivo anti-tumour activity (in Figure 3 and suppl figure 1) is most likely due to the altered NK cell development in these mice. This is true for both the Vav-Cre and Ncr1-Cre models. The authors must analyse degranulation, perforin and granzyme for the individual NK cell subsets (CD11b+CD27-, CD11b+CD27+, CD11b+CD27-). Killing assays should be performed on sorted NK cell subsets. These experiments are required to determine whether mTORC2 has a cell-intrinsic effect on NK cell cytotoxicity

Why this has not been satisfactorily addressed:

In the methods, the authors describe a cytotoxicity assay but they have not performed an NK cell cytotoxicity assay. They have just looked a degranulation which is different. Whether the NK cells are killing the target cells is not known. There are cytotoxicity assays that do not require Cr51 radioactivity such as using the fluorescent dye Calcein AM. The authors should demonstrate altered cytotoxicity.

Original comment:

Reviewer #2-2. Why do the authors use Rictor^{fl/+}/Ncr1-Cre⁺ and Rictor^{fl/fl}/Ncr1-Cre⁻ mice as

controls for *Rictor^{fl/fl}/Ncr1-Cre⁺* experimental mice? The correct controls should be *Rictor^{+/+}/Ncr1+Cre*. This is important because the *Ncr1* cre knockin mouse strain generated by Eric Vivier results in decreased expression of the *Ncr1*/NKp46 receptor. That said it is not clear where the *Ncr1* Cre used in this study came from; who generated and characterised them? This must be clarified

Why this has not been satisfactorily addressed:

The authors now use *Rictor^{fl/+}/Ncr1-Cre⁺* and *Raptor^{fl/+}/Ncr1-Cre⁺* mice as controls. So they are essentially looking at raptor hets versus raptor KO and rictor hets versus rictor KO. Why are they doing this? They should be comparing *Rictor^{fl/fl}/Ncr1-Cre⁺* (rictor KO) and *Rictor^{+/+}/Ncr1+Cre* (*Rictor* WT) and the same for raptor. The rationale behind using these mice as controls is not clear. I do not consider these appropriate control mice.

Original comment:

Reviewer #2-5. In figure 5 the authors explore the signalling relationship between mTORC1 and mTORC2. They show increased mTORC1 signalling in *Rictor* KO NK cells. Given that both mTORC1 and mTORC2 share the catalytic mTOR kinase subunit, this result could be explained by the fact that deletion of *Rictor* will release mTOR kinase subunits to form more mTORC1 complexes. Therefore, the authors need to demonstrate that the observed increases in mTORC1 signalling are an artefact of *Rictor* deletion. This is essential in order to make the conclusion that mTORC2 is a negative regulator of mTORC1 signalling in NK cells.

Why this has not been satisfactorily addressed:

The authors do not satisfactorily address this point. They now show increased pSTAT5 *Rictor* KO NK cells (something that was not observed by the recent study by Yang et al 2018 in eLife). However, they do not prove that this increase in STAT5 is causing the increased mTORC1 activity. They use a STAT5 inhibitor and show decreased mTORC1 signalling but this inhibitor will also inhibit the basal STAT5 signalling and not just the elevated pSTAT5, and STAT5 is important for the expression of *Slc7a5* in lymphocytes, which has recently been shown to be essential for mTORC1 activity in NK cells (Loftus et al, Nature Communications 2018). So it is not really unexpected to see reduced mTORC1 with a STAT5 inhibitor but these experiments do not prove that the increased mTORC1 in *Rictor* KOs is due to increased STAT5 activity.

The authors suggest that Akt is not important in the increased mTORC1 activity in *Rictor* knockouts and show equivalent pAkt308, but they do not do the simple experiment that would prove that Akt is required for mTORC1. They should treat the cells with Akt-1/2i. Loftus et al also showed that mTORC1 was not regulated by Akt in IL2 stimulated NK cells. Akt-1/2i is not toxic towards NK cells (as suggested by the authors) if used at an appropriate concentration (Loftus et al. Nat Commun 2018). Therefore, the authors do not provide any clear evidence of the mechanism linking *Rictor* deficiency to enhanced mTORC1 activity. It is still very possible that the increased mTORC1 is an artefact because deletion of *Rictor* is releasing mTOR kinase subunits to form more mTORC1 complexes.

Responses to Reviewers' Comments

Reviewer #1 (Remarks to the Author):

The authors' reasoning, that conditional knock-outs driven by Ncr1-iCre are usually not complete, is not satisfactory and brings up a host of questions that need to be addressed for proper conclusions to be drawn:

1) Certain subsets of developing NK cells may be more/less affected by protein loss, we are looking at the ones that managed to survive. Thus the phenotype may be a consequence of this surviving subset and not the phenotype of all NK cells lacking this protein.

Author response: We appreciate the concerns raised by the reviewer. To address this concern, we detected the viability of Rictor or Raptor deficient NK cells by 7-AAD/Annexin V staining assay. It turned out that neither Raptor nor Rictor deficiency could induce any discernible apoptosis in the different subsets of NK cells from spleen or BM. Alternatively, Raptor deficiency can protect NK cells from apoptosis in BM to some extent (please see Fig.R1 Q1). Based on the above evidence, we believe that the phenotype in our studies is indeed from NK cells lacking the proteins rather than just from the surviving subsets.

Fig. R1 Q1

Viable, 7-AAD⁻Annexin V⁻; Early apoptosis, 7-AAD⁻Annexin V⁺; Late apoptosis, 7-AAD⁺Annexin V⁺

2) At the very least the authors need to amend all their statements and conclusions such that they reflect that 'these are the phenotypes' in presence of reduced levels of the protein, not complete absence.

Author response: Thanks for the suggestion, we have made some statements to make our conclusion more precise in the Discussion section in our manuscript as followings (Line 387-390):

As Raptor or Rictor proteins were not completely deleted from NK cells in the Ncr1-Cre mice models, strictly speaking, the phenotypes we observed actually resulted from reduced expression but not complete loss of the target protein.

Reviewer #2 (Remarks to the Author):

The authors have satisfactorily addresses comment (1).

With respect to comment (2) I agree with the authors that while the controls used are not ideal it is unlikely that this affects the conclusions of this study.

With respect to comment (3), the key piece of data that they have generated is that of Slc7a5 expression in Rictor KO NK cells. However, historically there have been no slc7a5 antibodies that have worked by flow cytometry analysis. Therefore, the author must include important controls to demonstrate that this flow based analysis of slc7a5 is reliable.

Does the expression levels using this flow based approach follow those described by Loftus et al 2018 as assessed by mRNA analysis.

(i) does slc7a5 signal increase in cytokine (IL2/12) stimulated murine NK cells and also is this blocked by the STAT5 inhibitor.

(ii) If you then withdraw IL2 from 1 day IL2/12 activated NK cells for 8 hours is slc7a5 expression reduced.

(iii) the authors see a very nice decrease in pS6 levels in WT NK cells treated with the STAT5 inhibitor. Do you get a corresponding decrease in slc7a5 expression.

An alternative approach (and preferable) would be to demonstrate increased slc7a5 expression by another method. This would provide confidence in this key piece of the mechanism. Here I would suggest looking at Slc7a5 mRNA or using a flow based method for measuring slc7a5 mediated uptake (i.e. slc7a5 activity) that was described in Nat Commun recently (PMID:29773791)

Author response: We appreciate the concerns raised by the reviewer. Our rabbit anti-SLC7A5/AF647 conjugated antibody (bs-10125R-AF647) was purchased commercially from Bioss ANTIBODIES and was used for cell staining following the standard protocols for flow cytometry (detailed in our Methods section) with a dilution of 1:50. The unconjugated form of this SLC7A5 antibody (bs-10125R) has been tested for the reactivity with mouse in a work published on *Science* previously

(PMID : 27609895) (please see Fig. R2Q1a from the paper of PMID 27609895: reproduced below for your convenience). Now we carried out flow cytometric tests for the anti-SLC7A5/AF647 conjugated antibody following the conditions described by Loftus et al 2018 (PMID : 29904050) as the referee suggested. In our flow cytometric data shown in Fig R2 Q1b-c, we also observed a significant increase of SLC7A5 protein level on NK cells after treatment with IL2+IL12 for 18h, and this increase was efficiently blocked by treatment with STAT5-IN (Fig R2 Q1b). Furthermore, when IL2 was withdrawn from 20h IL2+IL12 activated NK cells for 8h, the SLC7A5 protein level also declined obviously (Fig R2 Q1c). These results are consistent with the analysis of SLC7A5 mRNA in the study of Loftus et al. (PMID : 29904050) and strongly support that our flow cytometric data with this anti-SLC7A5/AF647 conjugated antibody is reliable. In addition, as shown in Fig. 7i (main figure) and Fig. R2 Q1a, the SLC7A5 expression on WT NK cells was obviously inhibited by STAT5-IN.

REDACTED

REVIEWERS' COMMENTS:

Reviewer #2 (Remarks to the Author):

The authors have satisfactorily addressed my concerns.